# Structural basis of semaphorin–plexin *cis* interaction

Daniel Rozbesky[1] iD, Marieke G Verhagen[2], Dimple Karia[1], Gergely N Nagy[1], Luis Alvarez[3], Ross A Robinson[1,†], Karl Harlos[1] iD, Sergi Padilla-Parra[1,3,‡,§], R Jeroen Pasterkamp[2,*] iD & Edith Yvonne Jones[1,**] iD

## Abstract

Semaphorin ligands interact with plexin receptors to contribute to functions in the development of myriad tissues including neurite guidance and synaptic organisation within the nervous system. Cell-attached semaphorins interact in *trans* with plexins on opposing cells, but also in *cis* on the same cell. The interplay between *trans* and *cis* interactions is crucial for the regulated development of complex neural circuitry, but the underlying molecular mechanisms are uncharacterised. We have discovered a distinct mode of interaction through which the *Drosophila* semaphorin Sema1b and mouse Sema6A mediate binding in *cis* to their cognate plexin receptors. Our high-resolution structural, biophysical and *in vitro* analyses demonstrate that monomeric semaphorins can mediate a distinctive plexin binding mode. These findings suggest the interplay between monomeric *vs* dimeric states has a hereto unappreciated role in semaphorin biology, providing a mechanism by which Sema6s may balance *cis* and *trans* functionalities.

**Keywords** axon guidance; cis interaction; plexin; semaphorin; semaphorin signalling
**Subject Categories** Neuroscience; Structural Biology
**The EMBO Journal (2020) 39: e102926**

## Introduction

Semaphorins and plexins are one of the classical cell guidance ligand-receptor families first characterised by their ability to steer axon growth cones in the developing nervous system (Kolodkin *et al*, 1992, 1993; Luo *et al*, 1993; Tamagnone *et al*, 1999). Beyond axon guidance, semaphorin–plexin signalling is implicated in a plethora of physiological functions including other aspects of neural development, angiogenesis, vascularisation, organogenesis and regulation of immune responses (Tran *et al*, 2007; Pasterkamp, 2012; Takamatsu & Kumanogoh, 2012). Conversely, deregulation of semaphorin–plexin signalling is associated with tumour progression and other diseases (Tamagnone, 2012). Exquisite control of the local level and biological consequence of signalling is characteristic of the semaphorin–plexin system and essential for many of its functions.

Semaphorins are secreted, transmembrane or GPI-anchored proteins (Kolodkin *et al*, 1993). Membrane-attached semaphorins and plexins commonly function through cell-to-cell *trans* interactions in which the semaphorin ligands and plexin receptors are presented on opposing cells. However, when ligand and receptor are present on the same cell surface there is potential for ligand-receptor binding in *cis* at the same plasma membrane. An increasing body of evidence points to the importance of *cis* interactions in the regulation of diverse cell guidance signalling systems (Seiradake *et al*, 2016). In the semaphorin–plexin signalling system, *cis* interactions were first described between class 6 semaphorins (Sema6s) and their cognate plexin class A (PlxnA) receptors. Studies in migrating granule cells suggest that binding of Sema6A and PlxnA2 in *cis* inhibits the binding of PlxnA2 by Sema6A in *trans* as the absence of Sema6A in *cis* causes over-activation of PlxnA2 (Renaud *et al*, 2008). The *cis* interaction of Sema6A-PlxnA2 has been further reported to be essential for proper development of lamina-restricted projection of hippocampal mossy fibres (Suto *et al*, 2007; Tawarayama *et al*, 2010). Finally, the inhibitory effect of *cis* interaction has been demonstrated between Sema6A and PlxnA4 (Haklai-Topper *et al*, 2010), and Sema6B and PlxnA2 (Andermatt *et al*, 2014). Contrary to these inhibition effects, the *cis* interaction between semaphorin SMP-1 and the PlxnA4 homolog, PLX-1, in *C. elegans* has been shown to result in plexin activation (Mizumoto & Shen, 2013). Similarly, mouse Sema5A signals through PlxnA2 co-expressed on hippocampal dentate granule cells to regulate synaptogenesis (Duan

1 Division of Structural Biology, Wellcome Centre for Human Genetics, University of Oxford, Oxford, UK
2 Department of Translational Neuroscience, UMC Utrecht Brain Center, University Medical Center Utrecht, Utrecht University, Utrecht, The Netherlands
3 Cellular Imaging, Wellcome Centre for Human Genetics, University of Oxford, Oxford, UK
  *Corresponding author. Tel: +31 88756 8831; E-mail: r.j.pasterkamp@umcutrecht.nl
  **Corresponding author. Tel: +44 1865 287547; E-mail: yvonne@strubi.ox.ac.uk
  †Present address: Immunocore Ltd, Abingdon, UK
  ‡Present address: Department of Infectious Diseases, Faculty of Life Sciences & Medicine, King's College London, London, UK
  §Present address: Randall Centre for Cell and Molecular Biophysics, King's College London, London, UK
  [The copyright line of this article was changed on 17 June 2020 after original online publication.]

*et al*, 2014). Perhaps the most exquisite interplay of semaphorin–plexin *cis* and *trans* interactions reported to date is that of Sema6A and PlxnA2 in the elaboration of dendritic arbors during retinal circuit assembly (Sun *et al*, 2013). Intriguingly, it has been suggested that the *cis* and *trans* interaction modes of semaphorins and plexins require distinct binding sites (Haklai-Topper *et al*, 2010; Perez-Branguli *et al*, 2016).

The first crystal structures of semaphorins revealed that the hallmark N-terminal sema domain is a seven-bladed β-propeller with a propensity to dimerise (Antipenko *et al*, 2003; Love *et al*, 2003) and the homodimeric architecture has long been reported as essential for semaphorins to function as repulsive guidance cues (Klostermann *et al*, 1998; Koppel & Raper, 1998). In a recent study, we have demonstrated that semaphorins can also form heterodimers and monomers, and thus, their architecture is not restricted to homodimers (Rozbesky *et al*, 2019). Plexins are type I transmembrane proteins containing an N-terminal sema domain followed by multiple PSI and IPT domains in their extracellular segment (Bork *et al*, 1999; Tamagnone *et al*, 1999). The plexin intracellular region has a distinctive GAP domain architecture (He *et al*, 2009; Tong *et al*, 2009; Bell *et al*, 2011; Wang *et al*, 2013), which structural and functional studies suggest is activated by dimerisation (He *et al*, 2009; Tong *et al*, 2009; Wang *et al*, 2012, 2013). Recent crystal structures of full-length mouse PlxnA ectodomains (comprising ten domains) revealed a ring-like overall shape, which is presumably orientated parallel to the plane of the plasma membrane at the cell surface (Kong *et al*, 2016). The ring-like structure is consistent with an observed PlxnA-to-PlxnA "head-to-stalk" *cis* interaction being able to maintain pre-ligand bound plexins in a clustered, but autoinhibited, state on the cell surface, presumably by favouring separation, and thus preventing spontaneous dimerisation, of the transmembrane and intracellular regions (Kong *et al*, 2016). The existence of inactive dimers of pre-ligand bound plexin is further supported by data from fluorescence cross-correlation spectroscopy experiments on mouse PlxnA4 (Marita *et al*, 2015). Crystal structures have been reported for complexes formed between semaphorin ectodomains and fragments comprising up to four of the N-terminal domains of the cognate plexin ectodomain. These semaphorin–plexin complexes all show a bivalent 2:2 architecture that comprises a semaphorin dimer interacting with two copies of the plexin consistent with receptor activation by ligand-mediated dimerisation, a conclusion supported by structure-guided biophysical and cell-based assays (Janssen *et al*, 2010; Liu *et al*, 2010; Nogi *et al*, 2010). In all semaphorin–plexin complexes analysed to date, the semaphorins and plexins bind in a head-to-head (semaphorin sema domain-to-plexin sema domain) orientation suitable for a *trans* interaction between ligands and receptors attached to opposing cell surfaces triggering receptor activation (Kong *et al*, 2016). No molecular interaction surfaces have been characterised in terms of their ability to mediate semaphorin–plexin binding modes in *cis*; thus, the structural basis and molecular mechanism(s) governing the divergent outcomes of *cis* and *trans* binding remain elusive.

The ectodomain of Sema6A forms a weak dimer with monomeric and dimeric forms present in solution (Janssen *et al*, 2010; Nogi *et al*, 2010). The interplay of monomeric and dimeric Sema6 at the plasma membrane is likely relevant to *cis* interactions with the cognate PlxnA receptors. Structural and biophysical analyses at high concentrations have provided detailed insight into the interaction of

dimeric Sema6A with PlxnA2; however, because of the monomer-dimer equilibrium, the binding properties of wild-type monomeric Sema6A have eluded direct analysis. In structural and biophysical studies of the *Drosophila* semaphorin system, we recently discovered a wild-type monomeric semaphorin, Sema1b (Rozbesky *et al*, 2019). This unexpected discovery provided us with a system in which we could dissect the interaction surfaces, and contributions to plexin binding in *cis*, of a semaphorin that is purely in the monomeric state.

The class 1 (Sema1a and Sema1b) and class 2 (Sema2a and Sema2b) *Drosophila* semaphorins are membrane-attached and secreted, respectively. Sema1a and Sema1b are most closely related to the mammalian class 6 semaphorins and interact with the sole *Drosophila* class A plexin, PlexA (Pasterkamp, 2012). In previous studies, we have shown that the secreted *Drosophila* semaphorins, Sema2a and Sema2b, and also the ectodomain of membrane-attached Sema1a$_{ecto}$ are disulphide-linked dimers. All three of these semaphorins contain an intermolecular sema-to-sema disulphide bridge. Conversely, we found the ectodomain of membrane-attached Sema1b$_{ecto}$ to be a monomer in solution due to an amino acid substitution in the intermolecular disulphide bridge at position 254 (Rozbesky *et al*, 2019). Here, we show that *Drosophila* Sema1b is a monomer on the cell surface and can interact in *cis* with PlexA. We further report two crystal structures of Sema1b complexed with the semaphorin-binding region of PlexA. The crystal structures, along with biophysical and cell-based assays, show that monomeric Sema1b binds PlexA at two independent binding sites. One interaction mode corresponds to the canonical head-to-head orientation described previously for semaphorin–plexin binding. The second mode uses an interactive surface on Sema1b that is occluded in dimeric semaphorins. We were able to demonstrate that this novel "side-on" binding mode perturbs the ring-like structure of the PlexA ectodomain. In cell collapse assays, we found that the side-on mode of monomeric Sema1b-PlexA binding in *cis* was sufficient to inhibit PlexA signalling by dimeric Sema1a binding *in trans*. In dorsal root ganglion neurons, we also confirmed that mouse Sema6A utilises the same molecular mechanism for *cis* interaction with its cognate plexin receptor as its *Drosophila* homolog, Sema1b. Based on our findings, we propose models for semaphorin–plexin *cis* interactions which incorporate a distinctive role for monomeric semaphorin binding in the regulation of plexin signalling.

# Results

### Sema1b is a monomer on the cell surface and fails to mediate PlexA dimerisation

We considered the oligomeric state of Sema1b on the membrane of live cells. COS-7 cells were transiently transfected with Sema1b-F254C-mClover (a mutant which provides Sema1a-like disulphide-linked dimer formation) or with the wild-type Sema1b-mClover. Both constructs encompassed the ectodomain followed by a native transmembrane segment, short cytoplasmic linker and the C-terminal fluorescent protein mClover. mClover is a monomeric bright yellow-green fluorescent protein commonly used for the analysis of dimerisation or protein–protein interactions in live cells (Lam *et al*, 2012). Using Number and Brightness analysis, we determined a molecular brightness ($\varepsilon$) in live cells, which is directly related to the

oligomeric state. Number and Brightness analysis is a fluorescence fluctuation spectroscopy technique to measure the average number and oligomeric state of labelled entities in each pixel of a stack of fluorescently labelled images (Digman *et al*, 2008). We have recently developed the method further by implementing a novel detrending algorithm to detect monomers and dimers in live cells (Nolan *et al*, 2017, 2018a; Iliopoulou *et al*, 2018) or *in vitro* (Nolan *et al*, 2018b). Here, we calculated the molecular brightness of Sema1b-F254C-mClover to be double that of the molecular brightness of Sema1b-mClover consistent with Sema1b-mClover molecules being present on the membrane of COS-7 cells as monomers (Fig EV1A and B).

Our previous studies have shown that although monomeric, Sema1b$_{ecto}$ maintains PlexA binding in the nanomolar range (Rozbesky *et al*, 2019). To investigate whether Sema1b$_{ecto}$ dimerises or clusters PlexA on live cell surfaces, we probed the molecular brightness of PlexA-mClover on the membrane of COS-7 cells before and after stimulation with purified wild-type Sema1b$_{ecto}$ or the disulphide-linked dimer Sema1b$_{ecto}$-F254C. The PlexA-mClover construct contained the ectodomain followed by a transmembrane segment and the C-terminal fluorescent protein mClover. The addition of Sema1b$_{ecto}$-F254C resulted in a significant $3.0 \pm 1.8$ fold increase of the average molecular brightness which is likely related to a change of the PlexA-mClover oligomeric state. Conversely, the addition of wild-type Sema1b$_{ecto}$ at the same concentration had no noticeable effect on the average molecular brightness ($1.2 \pm 0.7$ fold increase) (Fig EV1C and D). Thus, though Sema1b binds PlexA in the nanomolar range, it fails to mediate PlexA dimerisation on the cell surface, presumably due to its monomeric state.

## A novel binding mode revealed by the crystal structure of the PlexA-Sema1b complex

We next determined crystal structures of the PlexA$_{1-4}$-Sema1b$_{1-2}$ complex from two different crystal forms (1:1 complex and 2:2 complex) at 3.0 and 4.8 Å resolution (Fig 1A–C, Table 1). The 1:1 complex crystal lattice contains one PlexA$_{1-4}$ monomer and one Sema1b$_{1-2}$ monomer per asymmetric unit. The crystal packing provides no Sema1b$_{1-2}$ dimerisation resembling that of the generic homodimeric architecture. The Sema1b$_{1-2}$ bound in the 1:1 complex with PlexA$_{1-4}$ is very similar to the unbound Sema1b$_{1-2}$ with the Cα rmsd of 0.81 Å indicating no large conformational changes upon complex formation. Only small differences in loop orientations at the ligand–receptor interface are apparent. The ectodomain of *Drosophila* PlexA has not been structurally characterised previously. The PlexA$_{1-4}$ structure in the 1:1 complex contains a sema domain composed of a seven-bladed β-propeller fold, which is followed by a PSI domain; however, we were not able to locate the IPT1-PSI2 domain segment. The sema domain of PlexA is most similar to mouse PlxnA2 with an rmsd of 1.43 Å over 424 matched Cα positions. In the 1:1 complex crystal structure, PlexA$_{1-4}$ and Sema1b$_{1-2}$ interact through their sema domains in a head-to-head orientation similar to the generic architecture shared by all reported structures of semaphorin–plexin complexes (Janssen *et al*, 2010; Liu *et al*, 2010; Nogi *et al*, 2010). The PlexA$_{1-4}$-Sema1b$_{1-2}$ interface buries a total solvent-accessible area of 1,837 Å$^2$. This extensive interface is composed of a mixture of hydrophobic and hydrophilic interactions similar to that of PlxnA2-Sema6A.

In the crystal of 2:2 complex, two pairs of 1:1 complexes are packed together in the asymmetric unit with a relative orientation of 168.8° to form a pseudo tetramer. Each Sema1b molecule binds to both PlexA molecules in the pseudo tetramer. There are therefore two independent interaction sites, A and B (Fig 1C), involving two different Sema1b-PlexA orientations. The first, head-to-head orientation is equivalent to that observed in the 1:1 complex (interaction site A; Fig 1B). In the second interaction mode, termed side-on (Fig 1D), Sema1b$_{1-2}$ and PlexA$_{1-4}$ bind through the site B or B' with their carboxy-terminal PSI domains oriented in parallel. The B and B' binding sites are not identical within the pseudo tetramer. While the B interaction site is extensive, the B' interaction site is formed through distant contacts between three residues only. Although the two PlexA molecules in the pseudo tetramer form a substantial interface, we did not observe any propensity for PlexA$_{1-4}$ to dimerise in solution to a concentration at least of 33 μM (Fig EV1E and F).

The structure of individual Sema1b$_{1-2}$ and PlexA$_{1-4}$ molecules in the 2:2 complex is very similar to those observed in the 1:1 complex, showing no significant conformational changes with the exception of the Exβ1-β2 loop in the extrusion of Sema1b$_{1-2}$, which adopts a different orientation in order to avoid steric clashes with PlexA$_{1-4}$. Unfortunately, we were not able to model the Exβ1-β2 loop completely because of fragmentary electron density; however, the loop's position is consistent with it making interactions to the PlexA$_{1-4}$. As well as undergoing this large-scale reorientation to accommodate PlexA$_{1-4}$, the Exβ1-β2 loop may also stabilise the complex.

In the side-on orientation, the sema domains of Sema1b$_{1-2}$ and PlexA$_{1-4}$ are bound in a configuration in which the bottom face of PlexA$_{1-4}$ is oriented perpendicularly to the side edge of Sema1b$_{1-2}$ (Fig 1D). The position of the B binding site between Sema1b and PlexA in the 2:2 complex is different to the binding site of the co-receptor neuropilin in the previously reported mouse Sema3A-PlexinA2-Nrp1 ternary complex (Janssen *et al*, 2012); however, they are positioned in very close proximity (Fig EV2A–C). Interface B can be divided into three main binding sites (Fig EV1G). The most prominent, site 1, is formed by the extrusion of Sema1b and blade 6 of PlexA. Site 2 is composed of the β4B-β4C loop of Sema1b and the β4D-β5A loop of PlexA. In site 3, the N-linked glycan at residue N289 of Sema1b forms contacts with the PSI1 domain of PlexA. The Exβ1-β2 and β4B-β4C loops are involved in semaphorin homodimerisation (Siebold & Jones, 2013); however, for Sema1b$_{1-2}$ in the 2:2 complex these loops mediate interaction with PlexA suggesting that a B-type interaction can only be mediated by a monomeric semaphorin molecule (Fig EV1H).

To confirm both interaction interfaces observed in the 2:2 complex, we produced three mutants of Sema1b$_{ecto}$, termed A, B or A+B, and analysed plexin binding using microscale thermophoresis (MST). In Sema1b-mutA, we mutated interface residues F203E, Q219R and K223E in order to disrupt the head-to-head interaction at site A. The single point mutations were designed to introduce electrostatic repulsions or reduction of surface hydrophobicity. Given the low resolution of the 2:2 complex and consequent lack of detailed information on residue-to-residue interactions, we decided to test the side-on interface by replacing each residue in the Exβ1-β2 loop by alanine rather than simple point mutations. Sema1b-mutA+B combined the modifications to potentially abolish both head-to-head and side-on interactions. All three Sema1b mutants were expressed and secreted at similar levels to the Sema1b wild-

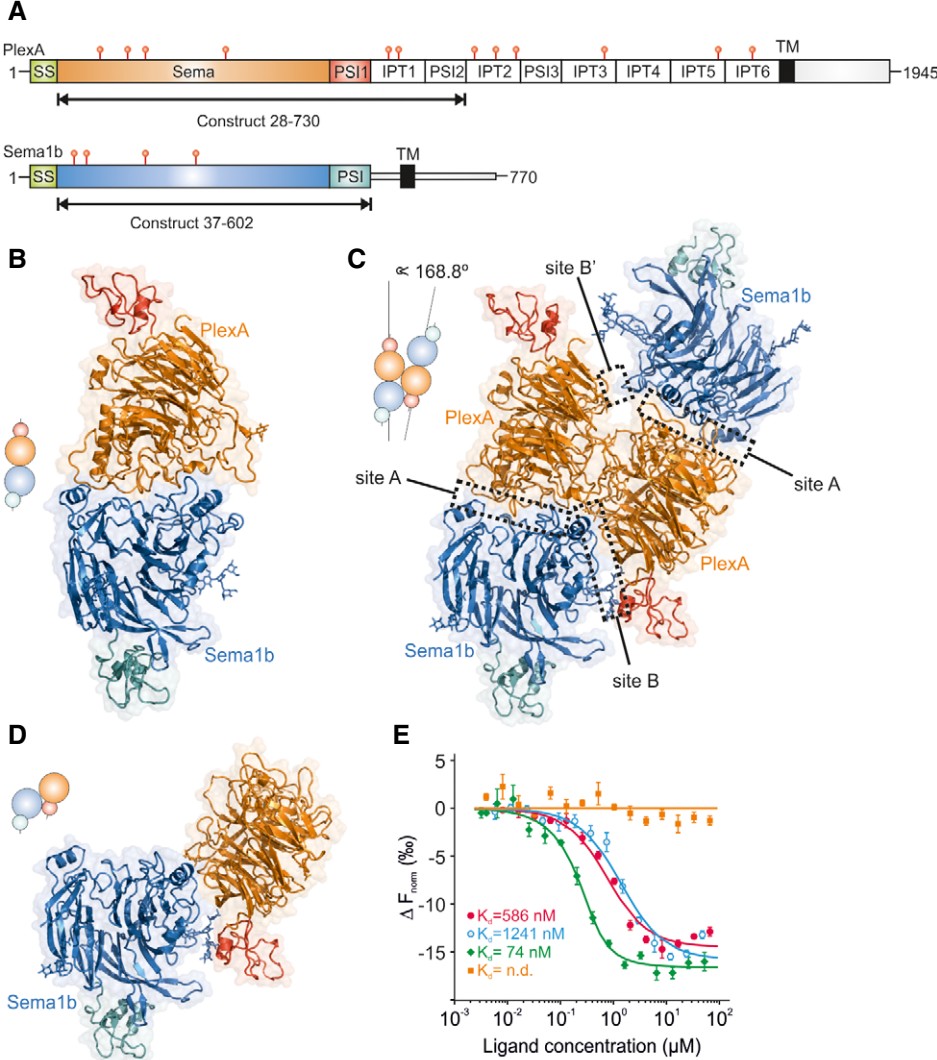

**Figure 1. Sema1b binds PlexA at two independent sites, A and B, indicating two independent Sema1b-PlexA orientations: head-to-head and side-on.**

A  Schematic domain organisation of *Drosophila* PlexA and Sema1b (SS, signal sequence; TM, transmembrane region, red symbols represent the position of N-linked glycosylation sites).

B, C  Ribbon representation of the $PlexA_{1-4}$-$Sema1b_{1-2}$ 1:1 (B) and 2:2 (C) complexes. N-glycans are shown in stick representation.

D  Ribbon representation of the side-on orientation derived from the 2:2 complex.

E  Microscale thermophoresis binding experiment for $PlexA_{1-4}$-mVenus and $Sema1b_{ecto}$ (red) or $Sema1b_{ecto}$-mutA (blue) or $Sema1b_{ecto}$-mutB (green,) or $Sema1b_{ecto}$-mutA+B (orange). The $Sema1b_{ecto}$ wild-type binding results are as reported previously (Rozbesky *et al*, 2019). All data were collected at the same time, and error bars represent s.d. of three technical replicates.

type. Furthermore, wild-type and all mutants were eluted at the same time from the size-exclusion column suggesting there is no problem with folding. We found using MST that both Sema1b-mutA and Sema1b-mutB maintained $PlexA_{1-4}$ binding while Sema1b-mutA+B did not provide any measurable indication of $PlexA_{1-4}$ binding at concentrations up to 66.3 μM (Figs 1E and EV2D–L). These data indicate that in solution $Sema1b_{ecto}$ can interact with $PlexA_{1-4}$ using either the head-to-head (site A) or side-on (site B) binding modes. We further assessed the stoichiometry of interaction between $Sema1b_{1-2}$ and $PlexA_{1-4}$ in solution using SEC-MALS. The unliganded $Sema1b_{1-2}$ or $PlexA_{1-4}$ eluted as a single peak corresponding to monomer (Fig EV2M and N); no propensity to dimerise was observed. SEC-MALS analysis of a sample containing an

equimolar mixture of $Sema1b_{1-2}$ and $PlexA_{1-4}$ revealed three peaks corresponding to unliganded $Sema1b_{1-2}$ and $PlexA_{1-4}$ and a $Sema1b_{1-2}$-$PlexA_{1-4}$ complex in 1:1 stoichiometry (Fig EV2O).

**Sema1b binds PlexA in *cis***

We then investigated whether Sema1b binds PlexA in *cis* on live cell surfaces by FRET-FLIM (Padilla-Parra *et al*, 2008; Padilla-Parra & Tramier, 2012; Kong *et al*, 2016). COS-7 cells were transiently co-transfected with PlexA-mClover and Sema1b-mRuby2. For cells co-expressing PlexA-mClover and Sema1b-mRuby2, we observed an average lifetime ($\tau_{av}$) of 2.51 ± 0.06 ns, while $\tau_{av}$ of 2.72 ± 0.03 ns was detected for cells expressing PlexA-mClover alone (Figs 2A and

**Table 1. Data collection and refinement statistics.**

| | PlexA$_{1-4}$-Sema1b$_{1-2}$1:1 complex | PlexA$_{1-4}$-Sema1b$_{1-2}$2:2 complex |
|---|---|---|
| Data collection | | |
| Space group | C 2 2 2$_1$ | P 6$_5$ 2 2 |
| Cell dimensions | | |
| a, b, c (Å) | 130.9, 195.1, 124.8 | 153.6, 153.6, 425.4 |
| α, β, γ (°) | 90, 90, 90 | 90, 90, 120 |
| Resolution (Å) | 76.86–2.96 (3.07–2.96) | 127–4.80 (4.97–4.80) |
| Unique reflections | 31,584 (2,791) | 15309 (1,485) |
| Multiplicity | 4.6 (2.2) | 20.9 (19.8) |
| Completeness (%) | 93.78 (83.79) | 99.90 (100.00) |
| I/σ(I) | 8.86 (1.75) | 7.41 (1.19) |
| Wilson B-factor (Å$^2$) | 69.89 | 217.35 |
| R-merge (%) | 17.4 (63.4) | 33.9 (224.1) |
| CC1/2 | 0.99 (0.548) | 0.998 (0.492) |
| CC[a] | 0.997 (0.841) | 1 (0.812) |
| Refinement | | |
| Resolution (Å) | 76.86–2.96 (3.06–2.96) | 127–4.80 (4.97–4.80) |
| Reflections used in refinement | 31582 | 15303 |
| Rwork/Rfree (%) | 18.62/24.63 | 28.66/30.68 |
| Number of atoms | 8,119 | 15,728 |
| Protein | 7,941 | 15,350 |
| Ligands | 178 | 378 |
| B-factor (Å$^2$) | | |
| Protein | 73.66 | 260.8 |
| Ligand | 129.39 | 297.52 |
| R.m.s. deviations | | |
| Bond lengths (Å) | 0.005 | 0.01 |
| Bond angles (°) | 0.76 | 1.43 |
| Ramachandran | | |
| Favoured (%) | 95.28 | 95.31 |
| Allowed (%) | 4.72 | 4.69 |
| Outliers (%) | 0 | 0 |

[a]Highest resolution shell is shown in parenthesis.

EV3A and B). The average fraction of the interacting donor (f$_D$) for cells co-expressing PlexA-mClover and Sema1b-mRuby2 was 0.26 ± 0.09 (Fig EV3C). This result clearly demonstrates that Sema1b indeed binds PlexA in *cis*.

Based on the architecture of our 2:2 complex, the *cis* interaction may be mediated by the head-to-head or side-on orientation between PlexA and Sema1b. Thus, to elucidate the structural basis for the PlexA-Sema1b *cis* interaction, we measured τ$_{av}$ for FRET-FLIM between PlexA-mClover and Sema1b-mRuby2 mutants, (site A, B and A+B mutants as described in the previous section). In order to assay the level of surface expression of the wild-type and mutant proteins, we measured fluorescence intensity on the membrane of COS-7 cells by TIRF microscopy. We observed comparable

intensities for all constructs (Fig EV3D and E) indicating that Sema1b wild-type and all three mutants are expressed at similar levels on the cell surface. As a negative control, we used COS-7 cells co-expressing PlexA-mClover and Sema1b-mutA+B-mRuby2 because Sema1b-mutA+B abolished both the head-to-head and side-on interactions in MST. These cells showed a τ$_{av}$ of 2.65 ± 0.03 ns, which is similar to the lifetime of donor alone with τ$_{av}$ of 2.72 ± 0.03 ns indicating no or very low FRET. A statistically significant (calculated by ANOVA test) shortening of the average lifetime due to FRET was observed for cells co-expressing PlexA-mClover and Sema1b-mutA-mRuby2 (τ$_{av}$ = 2.43 ± 0.10 ns) as well as for cells co-expressing PlexA-mClover and Sema1b-mutB-mRuby2 (τ$_{av}$ = 2.53 ± 0.04 ns) indicating that both head-to-head and side-on orientations can mediate *cis* interactions (Fig 2A). The side-on orientation appears to be more populated than the head-to-head as the average fraction of the interacting donor for cells co-expressing PlexA-mClover and Sema1b-mutA-mRuby2 (f$_D$ = 0.37 ± 0.15) is higher than for cells co-expressing PlexA-mClover and Sema1b-mutB-mRuby2 (f$_D$ = 0.21 ± 0.07) (Fig EV3C). Intriguingly, FRET-FLIM data indicate that the side-on orientation dominates over the canonical head-to-head orientation in the Sema1b-PlexA *cis* interaction on the cell surface. This observation appears to be counterintuitive to the structural data suggesting the side-on interface is weaker than the head-to-head interface. One possible explanation might be that partial deglycosylation of Sema1b$_{1-2}$ weakened the interaction mediated by the N-linked glycan at residue N289 of Sema1b in the 2:2 complex.

We then interrogated whether *cis* interaction occurs between PlexA and dimeric Sema1a, which has also been shown to bind PlexA (Winberg *et al*, 1998). Cells co-expressing PlexA-mClover and Sema1a-mRuby2 showed τ$_{av}$ of 2.70 ± 0.03 ns, which is similar to the lifetime of donor alone indicating no *cis* interaction (Fig 2A). Sema1a is tethered, from PSI domain to the membrane, by a linker that is some 35 residues shorter than in Sema1b, which is possibly insufficient to allow site A (head-to-head) mediated binding in *cis*. The lack of the PlexA-Sema1a *cis* interaction through an interaction site B on Sema1a is consistent with the inaccessibility of the sema domain loop Exβ1-β2 in dimeric semaphorins. Indeed, in all previously reported semaphorin crystal structures apart from Sema7A and the viral semaphorin A39R, the Exβ1-β2 loop is involved in the homodimerisation and is therefore not accessible for the side-on interaction with plexin (Fig EV3F). Thus, semaphorin–plexin binding through site B appears to be a unique feature of monomeric semaphorin molecules.

## PlexA ectodomain architecture and interactions

How does site B mediated interaction sit in the context of the structure and binding characteristics of full-length PlexA ectodomain? Recently, it has been shown that the ectodomains of human and mouse PlxnAs can adopt two distinct conformations: a preferred ring-like conformation and a less frequent chair-like conformation. In the ring-like conformation, the ectodomain forms a ring, which is nearly or fully closed by an intramolecular head-to-tail interaction (Kong *et al*, 2016; Suzuki *et al*, 2016). In light of the mammalian PlxnA ring-like conformation, we set out to assess the possible conformations of the *Drosophila* PlexA full ectodomain (PlexA$_{ecto}$) using negative stain EM. In the micrographs, PlexA$_{ecto}$ was monomeric. Single particle 2D class averages showed PlexA in the

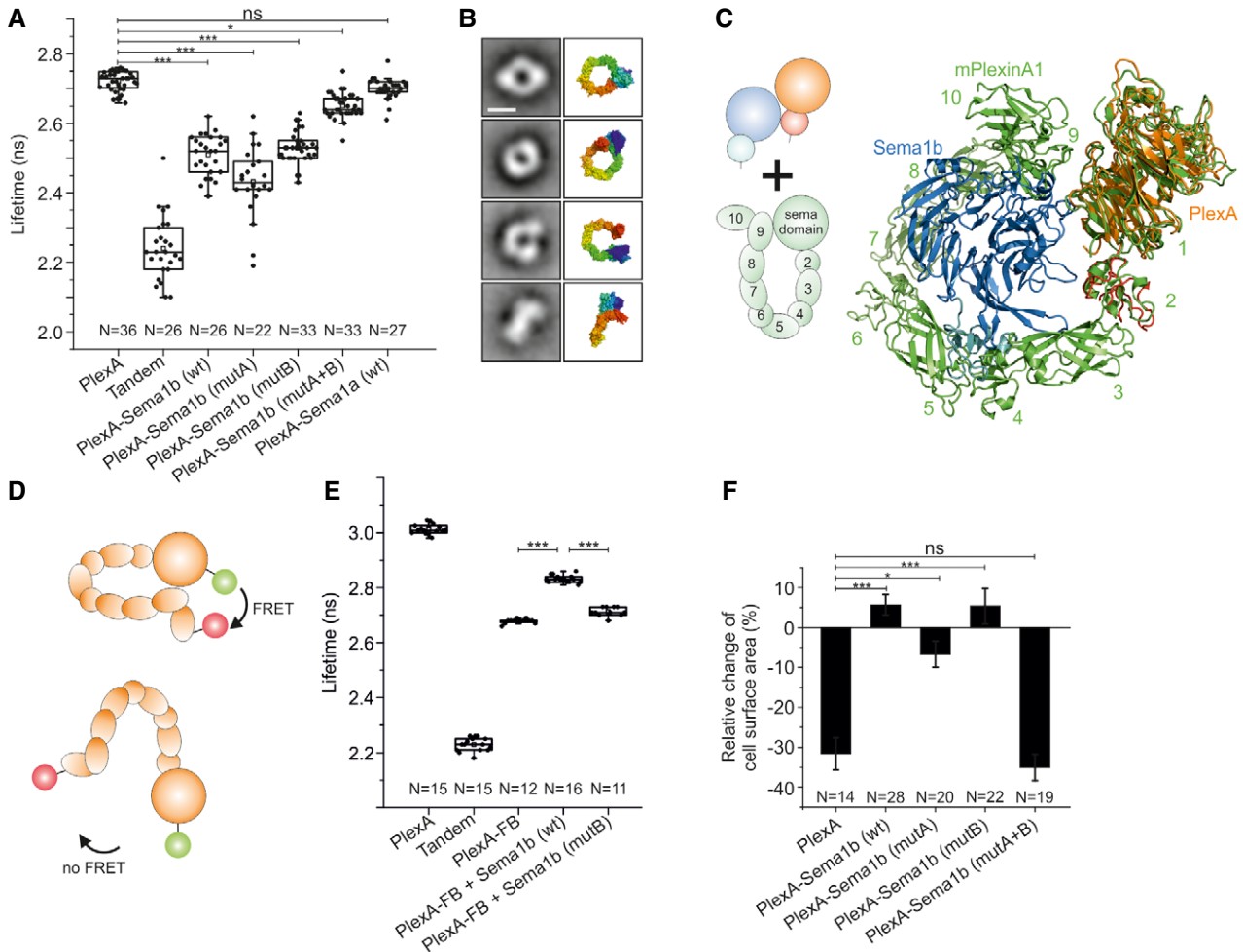

**Figure 2. Binding of Sema1b to PlexA in *cis* perturbs the PlexA ring-like conformation.**

A  FRET-FLIM in live COS-7 cells shows cell surface PlexA-Sema1b *cis* interaction. Sema1b mutants reveal that both head-to-head and side-on orientations of the PlexA-Sema1b complex are involved in the *cis* interaction. The *cis* interaction between Sema1a and PlexA is not observed. The boxcharts represent the average lifetime. Box limits indicate the 25th and 75th percentiles, centred lines show the median, squares represent sample means, whiskers extend 1.5-fold the interquartile range from the 25th and 75th percentiles, and *P*-value was calculated by one-way analysis of variance (ANOVA). *$P < 0.05$, ***$P < 0.001$.

B  Representative negative stain class averages of PlexA$_{ecto}$ correlate well with the ring-like conformation of mouse PlxnA1. The ring-like conformation of mouse PlxnA1 is shown in surface representation rainbow colour ramped from blue (N-terminus) to red (C-terminus). Scale bar, 10 nm. All class averages of PlexA$_{ecto}$ are shown in Fig EV3H.

C  Superposition of the side-on orientation derived from the 2:2 complex with the mouse PlexinA1 (pdb 5l56) ring structure (green) based on the *Drosophila* and mouse PlexinA1 sema domains.

D  Design of PlexA$_{ecto}$-FRET Biosensor for monitoring conformational changes. In the ring-like conformation, the fluorescent protein mClover and mRuby2 at N- and C- termini, respectively, are in close proximity generating FRET. Opening the ring-like conformation results in a FRET decrease.

E  *In vitro* FRET-FLIM measurement of the PlexA-FRET Biosensor shows that the PlexA ectodomain undergoes a conformational change upon treatment with Sema1b resulting in more open conformation. The boxcharts represent the average lifetime. Box limits indicate the 25th and 75th percentiles, centred lines show the median, squares represent sample means, whiskers extend 1.5-fold the interquartile range from the 25th and 75th percentiles, *P*-value was calculated by one-way analysis of variance (ANOVA). ***$P < 0.001$.

F  Relative change of COS-7 cells surface area upon treatment with Sema1a-Fc. Cells expressing PlexA$_{FL}$-mClover or PlexA$_{FL}$-mCover and Sema1b-mRuby2 wild-type or mutants were treated with purified Sema1a-Fc at a final concentration of 5.8 μM. Images were acquired every minute for 30 min. Cell surface area was calculated using ImageJ before and after stimulation. Data are presented as means ± sem. *$P < 0.05$, ***$P < 0.001$, *P*-value calculated by one-way analysis of variance (ANOVA).

ring-like conformation with overall shapes ranging from the nearly closed to the predominant, fully closed ring, which matches the mouse PlxnA ectodomain crystal structures and major 2D class averages (Figs 2B and EV3G and H).

The intermolecular head-to-stalk interaction previously reported to occur for mouse PlxnA ectodomains involves sema domain and

domain 4 and 5 residues that are conserved across the A class plexins in vertebrates (Kong *et al*, 2016). This conservation extends to *Drosophila* PlexA. We therefore used analytical ultracentrifugation sedimentation velocity experiments to examine whether *Drosophila* PlexA$_{ecto}$ forms a dimer in solution. As seen in the equivalent experiments for mouse PlxnA1 (Kong *et al*, 2016), *Drosophila* PlexA$_{ecto}$

exists in a heterogeneous mixture encompassing monomer up to tetramer (Fig EV3I). Since *Drosophila* PlexA$_{1-4}$ is a monomer in solution (Fig EV1E and F), these data suggest that, similar to mammalian class A plexins, PlexA$_{ecto}$ can form an intermolecular head-to-stalk interaction on the cell membrane to provide pre-ligand bound autoinhibition.

### PlexA-Sema1b *cis* interaction leads to opening of the PlexA ring-like conformation

In the mouse PlxnA1$_{1-10}$ ring-like conformation, the ring is closed by the intramolecular head-to-tail interaction between the sema domain (domain 1) and IPT5 domain (domain 9) (Kong *et al*, 2016). Structural superposition of the mouse PlxnA1$_{1-10}$ ring-like crystal structure and *Drosophila* PlexA$_{1-4}$ in the 2:2 complex (Fig 2C) revealed that the position of the head-to-tail interaction site in the mouse PlxnA1$_{1-10}$ is very close to that used for the side-on interaction in the 2:2 complex between PlexA and Sema1b. Sema1b-PlexA interaction through site B binding appears sterically incompatible with the PlexA ectodomain maintaining a fully closed ring-like conformation. This observation suggests that a side-on interaction with monomeric semaphorin in *cis* may provide a mechanism for opening the plexin ectodomain ring.

We hypothesised that Sema1b and the IPT5 domain of PlexA compete for PlexA sema domain binding. Thus interaction between PlexA and Sema1b in *cis* might move the IPT5 domain out to open the ring and make the binding site on the PlexA sema domain accessible for the side-on interaction with Sema1b. To test our hypothesis, we constructed a PlexA-FRET-Biosensor (PlexA-FB) containing the PlexA$_{1-10}$ ectodomain fused with the fluorescent proteins mClover and mRuby2 at N- and C-termini, respectively. In PlexA-FB, ring opening of the PlexA$_{1-10}$ ectodomain would lead to a change of FRET efficiency between donor and acceptor (Fig 2D). Apart from PlexA-FB, we expressed and purified PlexA-mClover and tandem mClover-mRuby2 protein as control samples. *In vitro* measurement of PlexA-mClover by FRET-FLIM revealed the average lifetime ($\tau_{av}$) of donor alone of $3.01 \pm 0.02$ ns. For the PlexA-FB, we observed a $\tau_{av}$ of $2.68 \pm 0.01$ ns indicating FRET consistent with mClover and mRuby2 being held in close proximity by the PlexA ring-like conformation (Fig 2E). The calculated apparent interfluorophore distance in the PlexA-FB was 80.8 Å assuming random interfluorophore orientation. Addition of Sema1b$_{ecto}$ led to an increase of $\tau_{av}$ to $2.83 \pm 0.01$ ns; however, this effect was not observed when using the Sema1b$_{ecto}$-mutantB that is unable to bind PlexA by side-on interaction. The increase of the average lifetime upon Sema1b$_{ecto}$ treatment is likely a result of lower FRET efficiency because, on average, the distance between the mClover and mRuby2 has lengthened. Assuming random interfluorophore orientation, the calculated apparent interfluorophore distance in the PlexA-FB was 90.2 Å upon Sema1b$_{ecto}$ treatment. These data are consistent with Sema1b side-on binding to PlexA perturbing the ring-like PlexA ectodomain to a more open conformation.

### PlexA-Sema1b *cis* interaction prevents *cis*-engaged PlexA from interacting with Sema1a in *trans*

*Drosophila* Sema1a has been shown to bind PlexA and their interaction in *trans* has been reported to be crucial for controlling axon guidance (Winberg *et al*, 1998). Therefore, we next investigated whether the PlexA-Sema1b *cis* interaction can serve as a competitive inhibitor for Sema1a binding in *trans*. We used a COS-7 cell-based assay as a heterologous system that can mimic growth cone collapse (Turner & Hall, 2006). COS-7 cells transiently expressing full-length PlexA$_{FL}$-mClover showed a typical well-spread morphology. Incubation of these cells with recombinant Sema1a-Fc induced a robust morphological collapse (Figs 2F and EV3J–O). Conversely, this effect was not observed when using COS-7 cells co-expressing PlexA$_{FL}$-mClover and Sema1b-mRuby2, indicating that dimeric Sema1a-Fc binding to PlexA in *trans* is blocked by the monovalent PlexA-Sema1b *cis* interaction. We also examined the ability of Sema1b mutants to inhibit collapse. Incubation with Sema1a-Fc did not significantly alter the morphology of COS-7 cells that co-expressed PlexA$_{FL}$-mClover and Sema1b-mutA-mRuby2 or Sema1b-mutB-mRuby2. Conversely, co-expression of PlexA$_{FL}$-mClover and Sema1b-mutA+B-mRuby2 resulted in COS-7 cells showing collapse on incubation with Sema1a-Fc similar to that observed for COS-7 cells expressing PlexA$_{FL}$-mClover alone. Taken together, these results show that the interaction of monomeric Sema1b with PlexA in *cis* can inhibit dimeric Sema1a signalling through PlexA in *trans*. Furthermore, our data support a model in which monovalent Sema1b-PlexA *cis* interaction can be mediated by two distinct binding sites, A and B, using head-to-head or side-on orientations, respectively.

### Mouse Sema6A and *Drosophila* Sema1b utilise the same head-to-head and side-on binding modes to interact with plexins in *cis*

We next addressed whether this model was of relevance to vertebrate semaphorin function. Based on our results for Sema1b, we designed three mutants of mouse Sema6A, termed A, B or A+B, and analysed their PlxnA2 binding using FRET-FLIM. Instead of point mutations, we introduced N-linked glycosylation sites at binding sites A and B, to provide substantial steric hindrance to interface formation. In Sema6A-mutA, we introduced the N-linked glycosylation site Sema6A H212N in order to disrupt the head-to-head interaction at site A. In Sema6A-mutB, we designed the N-linked glycosylation site Sema6A E345N, K347T to target the putative side-on interaction through site B. Sema6A-mutA+B combined both N-linked glycosylation sites to potentially abolish both head-to-head and side-on interactions.

For FRET-FLIM, we used mouse PlxnA2 fused with mClover while Sema6A wild-type or mutants were fused with mRuby2. We observed statistically significant (calculated by ANOVA test) shortening of the average lifetime due to FRET for cells co-expressing mouse PlxnA2 and Sema6A wild-type or Sema6A-mutA or Sema6A-mutB (Fig EV4A and B). Cells co-expressing mouse PlxnA2 and Sema6A-mutaA+B revealed a lifetime statistically similar to the lifetime of donor alone. Sema6A and its three mutants showed similar fluorescence intensities as measured in TIRF microscopy (Fig EV4C and D) confirming that they had comparable expression levels on the surface of live COS7 cells. Consistent with our findings for *Drosophila* Sema1b, these data suggest that mouse Sema6A can bind PlxnA2 in *cis* using both head-to-head and side-on interaction. For mouse PlxnA2 and Sema6A, we did not observe such dramatic changes in the average lifetime as those observed for their *Drosophila* counterparts, presumably because Sema6A potentially exists

on the cell surface as a mixture of monomers and non-covalent dimers while Sema1b is present exclusively as the monomer.

### *Cis* interaction between mouse Sema6A and PlxnA4 serves as an inhibitory mechanism to signalling in *trans*

We further investigated whether mouse Sema6A and *Drosophila* Sema1b utilise the same molecular mechanism to inhibit plexin function using a growth cone collapse assay. Here, we used dorsal root ganglion (DRG) neurons, which have been previously reported to endogenously express Sema6A and PlxnA4 (Suto *et al*, 2005; Haklai-Topper *et al*, 2010). Further, binding of Sema6A to PlxnA4 *in cis* has been shown to inhibit growth cone collapse induced by Sema6A presented *in trans* in DRG neurons (Haklai-Topper *et al*, 2010). We transfected cultured DRG neurons from *Sema6A* knockout embryos with Sema6A-EGFP wild-type or Sema6A-EGFP mutants (mutA, mutB or mutA+B) or EGFP alone. Three days post-transfection, DRG neurons were treated with purified Sema6A-Fc, and growth cone collapse was scaled from one (uncollapsed) to eight (fully collapsed) using a previously established growth cone morphology matrix (Fig 3A and B) (van Erp *et al*, 2015; Kong *et al*, 2016).

In a pilot study, GFP control transfection and treatment with different concentrations of Sema6A-Fc were tested. Following treatment with 75 nM purified Sema6A-Fc, we observed a robust growth cone collapse for cells expressing EGFP, as compared to a low level of collapse in control-treated growth cones (0 nM; Fig EV5). In contrast, *Sema6A* knockout cells transfected with a Sema6A wild-type construct were unresponsive to addition of purified Sema6A-Fc (at 1, 10 and 75 nM) indicating that interaction of Sema6A with its PlxnA4 receptor in *cis* serves as a competitive inhibitor for Sema6A-Fc binding in *trans*. A similar effect was observed for cells expressing Sema6A-mutA or Sema6A-mutB suggesting that both mutants maintained PlxnA4 binding in *cis* using the side-on or head-to-head sites, respectively. Conversely, cells expressing Sema6A-mutA+B showed modest (10 nM) or robust (75 nM) growth cone collapse similar to that observed for cells expressing EGFP alone (as shown for 75 nM). On basis of these data, a larger number of growth cones was analysed in 3 independent experiments using 75 nM Sema6A-Fc (Fig 3A and B). This analysis confirmed the pilot data showing strong Sema6A-Fc-induced growth cone collapse in EGFP-transfected Sema6A knock-out neurons, a rescue effect by transfection of Sema6A wild-type, and a failure to rescue by Sema6A-mutA+B. These findings are in agreement with the Sema1b cell-based assay. Overall, these results support our model that Sema6A-PlxnA4 *cis* interaction can be mediated by two distinct binding sites and also demonstrate that Sema6A-PlxnA4 *cis* interaction directly inhibits the plexin receptor's ability to respond to ligand binding in *trans*.

## Discussion

In the immune system, *cis* interactions provide a mechanism to fine tune the level of signalling at which a biological response is triggered (Held & Mariuzza, 2008). Similarly, *cis* interactions have been proposed to act as threshold-generating mechanisms for semaphorin-plexin, ephrin–Eph and Notch-Delta signalling during the development of the nervous system (Yaron & Sprinzak, 2012). For semaphorin–plexin signalling, there is particular abundance of evidence for *cis* interactions between vertebrate class 6 semaphorins and their plexin A receptors. Inhibitory *cis* interactions between Sema6s and PlxnAs have been reported to modulate repulsive cell guidance signalling in a number of neuronal cell types: dorsal root ganglion neurons, spinal cord, starburst amacrine cells in the retina and granule cell axons (mossy fibres) in the hippocampus (Suto *et al*, 2007; Renaud *et al*, 2008; Haklai-Topper *et al*, 2010; Tawarayama *et al*, 2010; Sun *et al*, 2013; Andermatt *et al*, 2014). In these examples, either *cis* interaction with a semaphorin ligand directly inhibits the plexin receptor's ability to respond to ligand binding in *trans* or plexin binding in *cis* sequesters the semaphorin ligand so that it cannot interact with a plexin receptor in *trans*. Both these scenarios require that the *cis* interaction between ligand and receptor does not activate the receptor. This poses a conundrum; how does the inhibitory interaction between ligand and receptor in *cis* differ from the activating interaction in *trans*? To date, research into the molecular mechanism underlying plexin activation by semaphorin binding in *trans* has highlighted the role of the dimeric ligand in cross-linking two receptors (Siebold & Jones, 2013; Jones, 2015). In this paper, we present structural, biophysical and cell-based analyses that delineate the distinctive properties of monomeric semaphorins interacting in *cis* with their plexin receptors. Our data suggest a mechanism by which cell-attached semaphorin molecules in the monomeric state can contribute to the inhibition of *trans* interactions through a side-on interaction in *cis*. This side-on interaction requires the semaphorin to be in the monomeric state, causes conformational change in the ectodomain of the plexin receptor and, independent of the well-established head-to-head semaphorin–plexin binding mode, can inhibit the activation of plexin receptor by dimeric semaphorin. Below we discuss these data

**Figure 3. Mouse Sema6A utilises the same molecular mechanism to inhibit plexin function as that observed for *Drosophila* Sema1b.**

A  Cultured dorsal root ganglion (DRG) neurons from E12.5 *Sema6A* knockout (KO) embryos were transfected with EGFP, EGFP-tagged wild-type (WT) Sema6A (Sema6A-WT) or different EGFP-tagged Sema6A mutants (Sema6A-mutA, Sema6A-mutB and Sema6A-mutA+B) at 1 day *in vitro* (DIV1) and treated with purified Sema6A-Fc at a concentration of 75 nM for 1 h at 37°C at DIV4. Transfection of EGFP was used as a control (full collapse). DRG neurons from *Sema6A* KO embryos acquired sensitivity to Sema6A and growth cone collapse was observed in three independent experiments. Transfection of Sema6A-WT, Sema6A-mutA and Sema6A-mutB but not of EGFP or Sema6A-mutantA+B prevented Sema6A-mediated growth cone collapse (Fisher's exact test *P* < 0.0001). Quantification of growth cone collapse was performed using a growth cone morphology matrix (lower part panel A). Data are presented as percentage of morphologically distinct growth cones, divided (represented by the dotted line) in two categories, from uncollapsed (1–4) to fully collapsed (5–8). Totally, 20–40 growth cones were analysed for each condition per experiment (*n* = 3 experiments). Scale bar, 20 μm.

B  Representative images of growth cones of mouse DRG neurons transfected with WT or mutant constructs treated with 75 nM Sema6A-Fc and visualised with immunofluorescent staining to detect GFP (green) and phalloidin (red). Average intensity images for all different constructs are shown in grey colour. Scale bar, 20 μm.

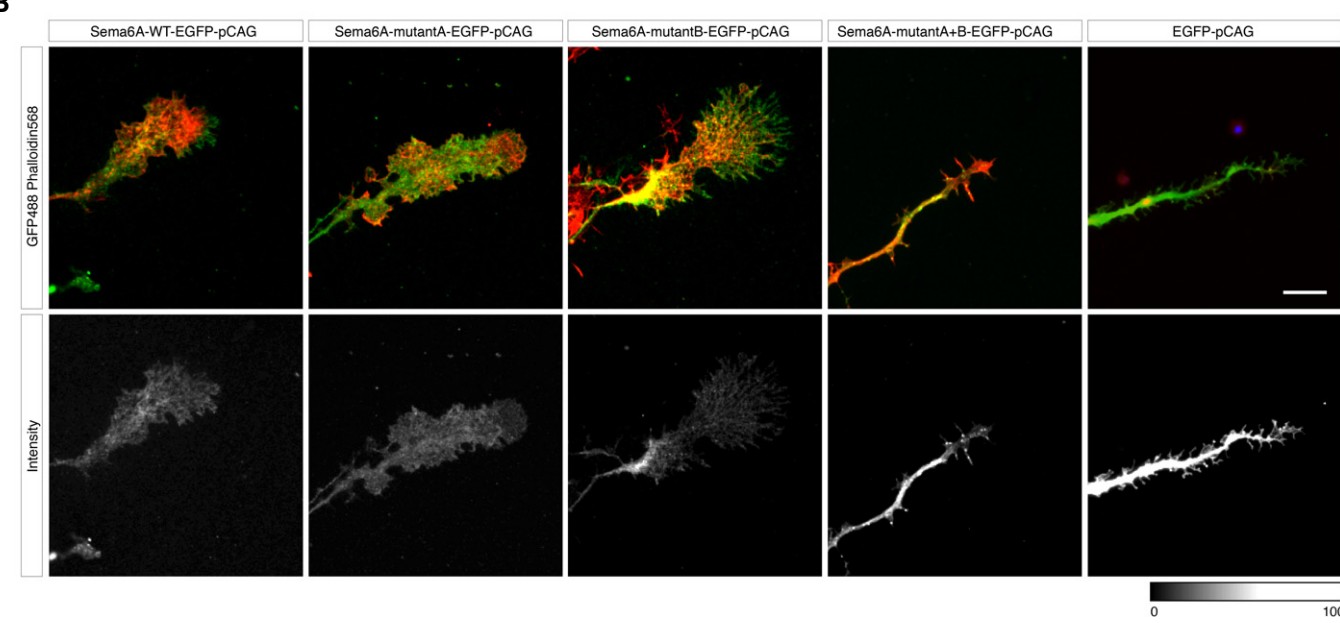

**Figure 3.**

and based on them propose molecular mechanisms that can provide distinct functional outcomes for *cis* and *trans* binding between a semaphorin ligand and plexin receptor.

### Monomeric semaphorins with distinctive side-on binding properties

We showed that *Drosophila* Sema1b exists on the cell surface in a monomeric state. Whereas *Drosophila* Sema1a, Sema2a and Sema2b can be locked into a dimeric state through formation of a sema-to-sema domain disulphide bond, Sema1b lacks this bond (Rozbesky *et al*, 2019). Some, but not all, of the mammalian semaphorins can form inter-chain disulphide bonds at various points in their ectodomains, however, we are not aware of any wild-type semaphorin that lacks a measureable propensity for sema-to-sema domain dimerisation other than Sema1b. In particular, the ectodomains of Sema6s, the vertebrate homologues of the class 1 semaphorins, do not form covalently stabilised dimers, and we and others have reported secreted forms of Sema6A to be in monomer-dimer equilibrium (Janssen *et al*, 2010; Nogi *et al*, 2010).

We analysed the atomic level determinants of Sema1b-PlexA binding in crystal structures of the complex to address whether the monomeric state of Sema1b per se provides this semaphorin ligand with distinctive properties promoting *cis* interaction. In addition to the head-to-head interaction, which for dimeric semaphorin binding has been shown to mediate receptor activation in *trans* (Janssen *et al*, 2010; Liu *et al*, 2010; Nogi *et al*, 2010), our X-ray crystallographic analysis revealed a putative side-on binding mode. This side-on binding mode only becomes possible if a semaphorin is in the monomeric state, because the interaction site is otherwise occluded by the dimer interface. Notably, all previously reported crystal structures of semaphorin–plexin complexes have involved semaphorin dimers, presumably because, even when a monomer-dimer equilibrium is present in solution, crystallisation favours the dimeric state. We were able to use structure-guided mutants of Sema1b and Sema6A in FRET-FLIM experiments to reveal that *cis* interactions between the semaphorins and their cognate plexins can be mediated by either the, more favoured, side-on orientation or, less frequently, the head-to-head orientation. These findings were supported by the results from collapse assays, which confirmed that both interaction modes were inhibitory. Thus, monomeric Sema1b and Sema6A can utilise a side-on mode of interaction to inhibit plexin activation. Interestingly, side-on *cis* interaction with plexin could also serve to sequester semaphorin from monomer-dimer equilibrium into an "inert" monomeric state, inhibiting dimeric engagement and activation of plexin in *trans*. This mechanism would be consistent with the observation that interaction of PlxnA2 and Sema6A in *cis* functions to inhibit activation of PlxnA4 in *trans* (Suto *et al*, 2007). In previous work, ectopic expression of Sema1b in muscle subsets revealed that Sema1b can act as a repulsive guidance cue (Winberg *et al*, 1998). Although elegant, a caveat of this study is that for this Sema1b gain-of-function experiment, Sema1b was expressed at high levels. It is possible that such high expression may lead to Sema1b dimerisation or multimerisation and thereby allow this semaphorin to act as a ligand. Evidence to support the idea that Sema1b acts as a repellent under physiological conditions is currently lacking.

### Mechanisms for inhibition in *cis* resulting from the monomeric side-on interaction

Previous studies on PlxnA4-Sema6A (Haklai-Topper *et al*, 2010) and PlxnA2-Sema6A (Perez-Branguli *et al*, 2016) have suggested that *cis* and *trans* interactions are mediated by two distinct modes. In Fig 4, we propose models for the molecular mechanisms underlying inhibitory semaphorin–plexin *cis* interactions that are consistent with the data we report in this paper.

First, we consider the likely conformation and interactive state of the PlexA in isolation at the cell surface. Our results show that, similar to PlxnA ectodomains (Kong *et al*, 2016), the PlexA ectodomain forms a ring-like structure and exists as a heterogeneous mixture of oligomeric states encompassing monomer up to tetramer in solution. As the PlexA ectodomain has high sequence similarity to the mammalian PlxnA ectodomains and shares the same structural and biophysical characteristics, it appears plausible that the PlexA ectodomain can also maintain a level of pre-ligand bound autoinhibition using the intermolecular head-to-stalk interaction reported for the PlxnAs (Kong *et al*, 2016). Bivalent ligand binding in *trans*

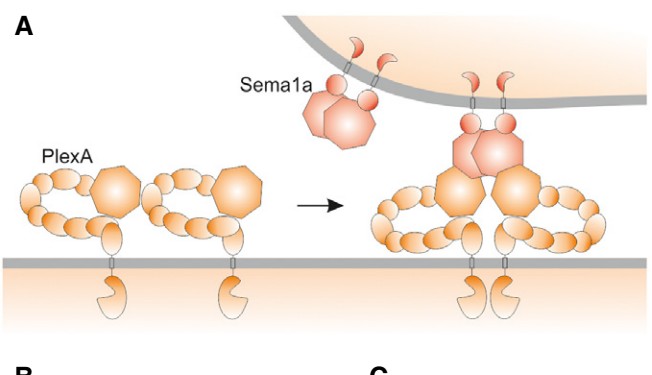

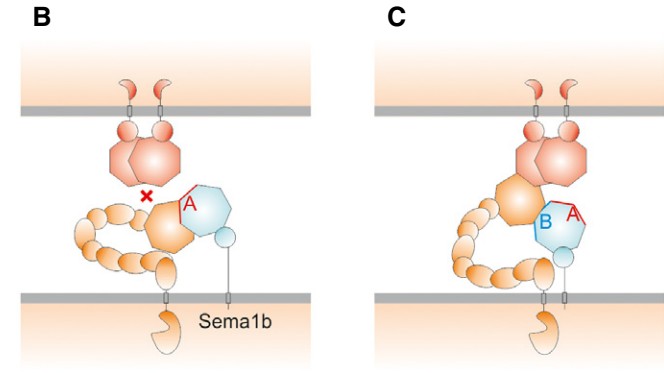

**Figure 4. Model for the PlexA-Sema1b *cis* interaction.**

A   The ring-like ectodomain of PlexA imposes a pre-signalling autoinhibition. Engagement of dimeric Sema1a with PlexA possibly disrupt the PlexA autoinhibitory dimer and switch it to the active dimer.
B   A structural model for the PlexA-Sema1b *cis* interaction. Monomeric Sema1b binds PlexA in the head-to-head orientation through site A without disruption of the PlexA ring-like.
C   In another model of the PlexA-Sema1b *cis* interaction, the monomeric state of Sema1b allows binding PlexA through the side-on binding site (site B) which may cause the PlexA ectodomain to be in a more open conformation. Possibly, the conformational change in the PlexA ectodomain makes the binding site on the PlexA more accessible for interaction with Sema1a presented in *trans*, however, Sema1a fails to dimerise PlexA because of steric clashes.

overcomes receptor autoinhibition. The *trans* interaction between PlexA and dimeric Sema1a can be expected to conform to the canonical semaphorin–plexin architecture resulting in dimerisation and activation of the PlexA (Fig 4A).

Inhibitory interactions between ligand and receptor in *cis* provide a mechanism for modulating the threshold for activation of signalling by the *trans* interaction. However, this modulatory mechanism requires that the ligand interact with the receptor without activating it. Our results suggest that this can be achieved in two ways. In mode A, the sema domain of monomeric Sema1b binds the sema domain of PlexA in the canonical head-to-head mode to directly inhibit head-to-head binding in *trans* (Fig 4B). This mode requires that the semaphorin ligand has sufficient flexibility in its ectodomain to position its sema domain at the requisite height and orientation relative to the plexin receptor. If a dimeric semaphorin is able to make mode A interactions with both its sema domains, this form of *cis* interaction could presumably serve to activate the plexin receptor, consistent with the reports of *cis* activation for the semaphorin SMP-1 and plexin PLX-1 in *C. elegans* (Mizumoto & Shen, 2013) and for mouse Sema5A signalling through PlxnA2 co-expressed on hippocampal dentate granule cells (Duan *et al*, 2014).

Mode B is the previously unobserved side-on interaction mode revealed in our studies of Sema1b. Interestingly, FRET-FLIM measurements suggest that this side-on interaction mode is more populated than the head-to-head *cis* interaction of mode A. The side-on interaction does not directly involve the head-to-head ligand binding site on PlexA, but it is inhibitory in cell collapse assays. We propose that side-on binding by monomeric semaphorin prevents formation of the 2:2 arrangement of semaphorin dimer complexed with two ring-like plexin ectodomains required to trigger a repulsive signal (Fig 4C). One subunit of the dimeric Sema1a could form a 2:1 complex with PlexA by head-to-head *trans* binding, but side-on Sema1b binding in *cis* would sterically hinder the engagement of a second PlexA in the orientation seen for canonical 2:2 complexes. Furthermore, our PlexA-FRET biosensor data suggest that *cis* binding of monomeric Sema1b to PlexA through the side-on interaction forces the ring-like structure of the PlexA ectodomain into more open conformation. It is possible that this altered PlexA ectodomain conformation may itself prevent activation of signalling for cell collapse.

Our studies on Sema1b have provided the first insights into the *cis* binding modes of monomeric semaphorins. Furthermore, our biophysical and cellular assays have demonstrated that Sema6A utilises the same molecular mechanism to occlude plexin function. The molecular mechanisms for semaphorin–plexin *cis* interaction we propose here suggest that the balance between monomeric and dimeric states is central to the biological functions of the Sema6s. Further investigation will be required to tease out how the interplay between monomeric and dimeric states of Sema6 ligands, expression levels of Sema6s and their PlxnA receptors, and *cis* and *trans* binding affinities in the context of cell membranes, combine to set signalling thresholds.

# Materials and Methods

## Protein production

Constructs encoding *Drosophila melanogaster* Sema1b$_{ecto}$, Sema1b$_{1-2}$, PlexA$_{1-4}$ and PlexA$_{ecto}$ (residues 37–659, 37–602, 28–730 and 28–

1,272, respectively) were cloned into pHLsec vector (Aricescu *et al*, 2006) in-frame with a C-terminal hexahistidine (His6) tag. A human IgGγ1 hinge and Fc-fusion construct of Sema1a (residues 21–602) or mouse Sema6A (residues 19–571) was constructed using pHL-FcHis vector (Aricescu *et al*, 2006). For MST experiments, PlexA$_{1-4}$ was cloned in frame with a C-terminal monoVenus (mVenus) followed by a C-terminal (His6) tag in pHLsec (Aricescu *et al*, 2006).

For crystallisation experiments, proteins were produced by transient transfection in HEK 293T cells (ATCC CRL-3216) in the presence of the α-mannosidase inhibitor kifunensine (Chang *et al*, 2007). For all other experiments, proteins were produced in HEK 293T cells without kifunensine. Five days post-transfection, the conditioned medium was collected and buffer exchanged using a QuixStand diafiltration system (GE Healthcare) and subjected to immobilised metal-affinity chromatography using a HisTrap FF column (GE Healthcare) and further purified by size-exclusion chromatography using a Superdex 200 16/60 column (GE Healthcare).

To investigate conformational changes of PlexA ectodomain, we produced a PlexA-FRET-Biosensor (PlexA-FB). First, based on pHLsec vector, we developed a novel pHLsec-FRET-Biosensor vector (pHLsec-FB), in which inserts can be cloned between AgeI and KpnI sites allowing the protein to be secreted with the fluorescent proteins mClover and mRuby2, at the N-terminus and the C-terminus, respectively, followed by the C-terminal octahistidine-tag. We produced PlexA-FB, PlexA-mClover and tandem mClover-mRuby2 by transient transfection in HEK 293T cells as described above. PlexA-mClover and tandem mClover-mRuby2 were used as control samples. For PlexA-FB, a construct encoding *Drosophila* PlexA$_{1-10}$ (residues 67–1,274) was cloned into pHLsec-FB. For the tandem mClover-mRuby2, an insert containing GGGGGA sequence encoding Gly-Gly was cloned into pHLsec-FB. For PlexA-mClover, a construct encoding PlexA$_{1-10}$ (residues 67–1,274) was C-terminally tagged with mClover followed by the C-terminal His6-tag.

## Site-directed mutagenesis

Site-directed mutagenesis of Sema1b was carried out by overlap-extension PCR and the resulting PCR products were cloned into the pHLsec vector as described above. In Sema1b-mutant A (mutA), we mutated interface residues F203E, Q219R and K223E. For Sema1b-mutant B (mutB), in which the loop 350–367 was replaced with 18 alanine residues, a synthetic clone was commercially synthesised (Invitrogen-GeneArt). Sema1b mutant A+B (mutA+B) combined the previous modifications. All mutant Sema1b proteins were secreted at similar levels to the wild-type protein.

## Crystallisation and data collection

Proteins for crystallisation were prepared in a buffer consisting of 15 mM Tris-HCl (pH 8.0) and 50 mM NaCl. For the Sema1b$_{1-2}$-PlexA$_{1-4}$ complex crystallisation, proteins were mixed at a 1:1 molar ratio and concentrated to 7.6 mg/ml. Prior to complex formation, Sema1b$_{1-2}$ was treated with endoglycosidase F1 (1:100 w/w) for 1 h at 37°C, whereas PlexA$_{1-4}$ was not deglycosylated. Sitting drop vapour diffusion crystallisation trials were set up using a Cartesian Technologies pipetting robot and consisted of 100 nl protein solution and 100 nl reservoir solution (Walter *et al*, 2005). Plates were maintained at 20°C in a Formulatrix storage

and imaging system. The Sema1b$_{1-2}$-PlexA$_{1-4}$ complex crystallised in two different crystal forms. One crystal form, with space group C2221, was grown in 0.1 M HEPES (pH 7.0) and 8% (w/v) PEG 8000, the other crystal form, with space group P6522, crystallised in 0.1 M MES (pH 6.5) and 12% (w/v) PEG 20,000. Crystals were cryoprotected by soaking in reservoir solution supplemented with 25% (v/v) glycerol and then flash-cooled in liquid nitrogen. Diffraction data were collected at 100K at Diamond Light Source beamlines I03 and I24 and indexed, integrated and scaled using the automated XIA2 expert system (Winter, 2010). For PlexA$_{1-4}$-Sema1b$_{1-2}$ 1:1 complex, due to radiation sensitivity, diffraction data from two separate isomorphic crystals were merged together to increase completeness.

### Structure determination and refinement

The structure of the PlexA$_{1-4}$-Sema1b$_{1-2}$ 1:1 complex was solved by molecular replacement in PHASER with the Sema1b$_{1-2}$ structure (PDB 6FKK) and the PlexinA2$_{1-4}$ structure (PDB 3OKY) (Janssen *et al*, 2010) as search models. This solution was re-built automatically by BUCCANEER and completed by several cycles of manual rebuilding in COOT and refinement in PHENIX. PlexA domains IPT1-PSI2 were omitted from the model due to disorder. The structure of the PlexA$_{1-4}$-Sema1b$_{1-2}$ 2:2 complex was solved by molecular replacement in PHASER with the Sema1b$_{1-2}$ and PlexA$_{1-2}$ structures as search models. This solution was refined by Rosetta (DiMaio *et al*, 2013) and rigid-body in PHENIX with each domain as a rigid group and a single *B* factor per domain and using global NCS restraints. Structure validation was performed using MolProbity (Davis *et al*, 2007). Refinement statistics are given in Table 1.

Buried surface areas of protein–protein interactions were calculated using PISA (Krissinel & Henrick, 2007), alignments were generated with Clustal Omega (Sievers *et al*, 2011), structural alignment was performed using PDBeFold (Krissinel & Henrick, 2004), and electrostatics potentials were generated using APBS (Baker *et al*, 2001). Figures were produced with PyMOL (Schrodinger, LLC), ESPRIT (Gouet *et al*, 1999) and Corel Draw (Corel Corporation).

### Microscale thermophoresis (MST)

MST experiments were performed using a Nanotemper Monolith NT.115 instrument (Nanotemper) at 28°C in 15 mM HEPES (pH 7.4), 150 mM NaCl, 2 mM CaCl$_2$ and 0.05% (v/v) Tween-20. A dilution series was prepared and a concentration of the fluorescent PlexA$_{1-4}$-mVenus was kept constant in all samples and the unlabelled Sema1b$_{ecto}$ mutants were varied in 1:1 dilution to give a titration. The samples were equilibrated 1 h at room temperature before filling into the standard capillaries (Nanotemper). To find the best thermophoretic setting, a measurement at 20, 40, 60 and 80% MST power was performed and the best signal to noise ratio was obtained by using 80% MST power. The LED power was set to 40%. The overall measurement time consisted of 5 s of cold fluorescence followed by IR-laser on and off times set at 30 and 5 s. Data were analysed using the MO Affinity Analysis v2.1.3 software (Nanotemper). The experiments were performed with three independent replicates.

### Analytical ultracentrifugation

Sedimentation velocity experiments were performed using an Optima XL-I analytical ultracentrifuge (Beckman). PlexA$_{1-4}$ and PlexA$_{ecto}$ samples in 15 mM HEPES (pH 7.4) and 150 mM NaCl were centrifuged in double sector 12 mm centerpieces in an An-60 Ti rotor (Beckman) at 40,000 rpm. Protein sedimentation was monitored by an absorption optical system and Rayleigh interference system. Data were analysed using SEDFIT (Schuck, 2000). Expected sedimentation coefficients of the structural models were predicted using WinHydroPRO (Ortega *et al*, 2011).

### Size-exclusion chromatography with multi-angle light scattering (SEC-MALS)

SEC-MALS experiments were carried out with the Superdex 200 10/300 column (GE Healthcare) connected online with a static light scattering (DAWN HELEOS II, Wyatt Technology), differential refractive index (Optilab rEX, Wyatt Technology) and Agilent 1200 UV (Agilent Technologies) detectors. PlexA$_{1-4}$ were injected into the column at flow rate of 0.5 ml/min in 15 mM HEPES (pH 7.8) and 150 mM NaCl. The molecular mass of glycoproteins containing N-linked oligomannose-type sugars was determined using an adapted RI increment value (dn/dc standard value, 0.185 ml/g). Data were analysed using the ASTRA software (Wyatt Technology).

### Single particle negative stain electron microscopy

A freshly purified PlexA$_{ecto}$ (3.5 μg/ml) was stained with 0.75% uranyl formate using the conventional negative staining protocol (Booth *et al*, 2011). Images were recorded using a Tecnai T12 transmission electron microscope operated at 120 kV on a 4,000 × 4,000 high-sensitivity FEI Eagle at magnification of 67,000, which corresponds to 1.68 Å/pixel sampling of the specimen. A defocus value of about −1.5 μm was used. Particles were manually selected and processed using the Eman2 (Tang *et al*, 2007) and Imagic (van Heel & Keegstra, 1981) software.

### Fluorescence resonance energy transfer—fluorescence lifetime imaging microscopy (FRET-FLIM) in live cells

*Drosophila melanogaster* PlexA (residues 28–1,311) or Sema1b (residues 37–686) and Sema1a (residues 21–633) were cloned into pHLsec vector in-frame with C-terminal fluorescent proteins mClover or mRuby2, respectively. For FRET-FLIM analysis of mouse Sema6A and PlxnA2, genes encoding mouse Sema6A (residues 19–675) and PlxnA2 (residues 36–1,263) were cloned into pHR-CMV-TetO2 vector in frame with C-terminal fluorescent proteins mClover and mRuby2, respectively (Elegheert *et al*, 2018). All constructs encompassed the ectodomain followed by a transmembrane segment and the C-terminal fluorescent protein. The same *Drosophila* constructs were used for Number and Brightness analysis. COS-7 cells (ATCC CRL-1651) grown on glass-bottom 35 mm Petri dishes (Mattek) were transiently co-transfected with *Drosophila* PlexA-mClover and Sema1b-mRuby2 or a donor-only sample (PlexA-mClover) or a fusion construct of mClover-mRuby2, which was used as a positive control. For mouse Sema6A and PlxnA2, we used lentiviral transduction of

COS-7 cells followed by FACS to enrich subpopulations of transduced cells (Elegheert *et al*, 2018).

Before imaging, a Dulbecco's modified eagle medium was replaced with PBS equilibrated at 37°C. Multicolour images were acquired 2 days post-transfection using a Leica SP8-X-SMD confocal microscope (Leica Microsystems) with a 63 × /1.40 numerical aperture oil immersion objective. mClover and mRuby2 were excited at 488 and 561 nm, respectively, and the fluorescence emission was detected using two hybrid detectors in photon counting mode at 498–551 and 573–625 nm, respectively.

FRET detection was based on the time domain FLIM experiments which were performed using a Time-Correlated Single Photon Counting (TCSPC) system operated by a PicoHarp 300 module (PicoQuant) attached to the Leica SP8-X-SMD confocal microscope (Leica Microsystems) with a 63 × /1.40 numerical aperture oil immersion objective at 37°C. A 488 nm picosecond pulsed diode laser PDL 800-B (PicoQuant) tuned at 80 MHz was used to excite the donor and the emitted photons passing through the 500–550 nm emission filter were detected using an external hybrid detector in photon counting mode. At least 600 photon events per pixel were collected in all cases and the lifetime analysis was carried out using a Symphotime (PicoQuant). The acquired fluorescent decays $i(t)$ were fitted by mono- equation (1) or bi-exponential equation (2) model.

$$i(t) = Ae^{-t/\tau_1} \tag{1}$$

$$i(t) = A_1 e^{-t/\tau_1} + A_2 e^{-t/\tau_2} \tag{2}$$

In equations (1) and (2), $\tau_1$ is the lifetime of the donor alone, $\tau_2$ is the lifetime of the donor in the presence of the acceptor, A, $A_1$ and $A_2$ are amplitudes. The average donor lifetime obtained from a mono-exponential fit from the cells expressing the donor only (PlexA-mClover) was fixed in the bi-exponential model to calculate the remaining two amplitudes and the second lifetime (Padilla-Parra *et al*, 2008; Padilla-Parra & Tramier, 2012). The amplitude weighted average lifetime of donor ($\tau_{av}$) was calculated using the equation equation (3):

$$\tau_{av} = \sum_i A_i \tau_i / \sum_i A_i \tag{3}$$

The fraction of the interacting donor ($f_D$) was calculated using the equation equation (4):

$$f_D = A_2 / (A_1 + A_2) \tag{4}$$

The fraction of the interacting donor was normalised by multiplying by a factor of 2 because we were able to detect just ~50% of the real interaction (Padilla-Parra *et al*, 2009).

### Fluorescence resonance energy transfer—fluorescence lifetime imaging microscopy (FRET-FLIM) in solution

For FLIM measurements *in vitro*, the purified fluorescent proteins were diluted to a concentration of 106 nM in 15 mM HEPES (pH 7.6) and 150 mM NaCl and 30 μl of the protein sample was loaded onto the μ-Slide with 18 wells (Ibidi). FRET was measured using the time domain FLIM experiments which were performed using a

Time-Correlated Single Photon Counting (TCSPC) system operated by a PicoHarp 300 module (PicoQuant) attached to the Leica SP8-X-SMD confocal microscope (Leica Microsystems) with a 63 × /1.40 numerical aperture oil immersion objective at room temperature.

A 488 nm picosecond pulsed diode laser PDL 800-B (PicoQuant) tuned at 40 MHz was used to excite the donor, and the emitted photons passing through the 500–550 emission filter were detected using an external hybrid detector in photon counting mode for a period of 350 s. The lifetime analysis was carried out using a Symphotime (PicoQuant). The acquired fluorescent decays were fitted by mono- or bi-exponential model as described above. The apparent interfluorophore distance $r$ was calculated from the equation equation (5):

$$E_{app} = 1 - \tau_{DA,av}/\tau_{D,av} = R_0^6 / (R_0^6 + r^6) \tag{5}$$

In equation (5), $\tau_D$ is the lifetime of the donor alone, $\tau_{DA}$ is the lifetime of the donor in the presence of the acceptor and $R_0$ is the Förster radius. The $R_0$ for the mClover/mRuby2 pair was calculated by the following equation

$$R_0 = \left( \frac{9,000 \ln 10 \, \Phi_D \kappa^2 J}{128 \pi^5 N_A n^4} \right)^{1/6} \tag{6}$$

Where $\Phi_D$ is the fluorescence quantum yield of the donor in the absence of the acceptor, $\kappa$ is the dipole orientation factor, $n$ is the refractive index of the medium, $N_A$ is Avogadro's number, and $J$ is the spectral overlap integral calculated as

$$J = \int F_D(\lambda)\varepsilon_A(\lambda)\lambda^4 d\lambda / \int F_D(\lambda)d\lambda \tag{7}$$

Where $F_D$ is the donor emission spectrum, and $\varepsilon_A$ is the acceptor molar extinction coefficient. To calculate the apparent interfluorophore distance, we used the Förster radius of 57 Å for the mClover-mRuby2 assuming random interfluorophore orientation. The spectral overlap integral of the mClover/mRuby2 pair was calculated using the Simpson method utilising the normalised excitation spectrum of the donor and the acceptor corrected for the published extinction coefficient (Lam *et al*, 2013). Knowing the value of J we calculated $R_0$ assuming random interfluorophore orientation ($\kappa^2 = 2/3$).

### Number & brightness analysis

COS-7 cells grown on μ-Slides (chambered coverslips) with 8 wells (Ibidi) were transiently transfected with *Drosophila melanogaster* PlexA-mClover (residues 28–1,311), Sema1b-mClover (residues 37–686) or Sema1b-F254C-mClover (residues 37–686). Before imaging, the cells were washed with PBS and Dulbecco's modified eagle medium was replaced with a phenol red-free Dulbecco's modified eagle medium equilibrated at 37°C.

Images were acquired using a Leica SP8-X-SMD confocal microscope (Leica Microsystems) with a 63 × /1.40 numerical aperture oil immersion objective at 37°C. For each studied cell, a single plane stack of 500 images was acquired at a resolution of 256 × 256 pixels and pixel size of 481 nm, with a pixel dwell time 2.43 μs. mClover was excited at 488 nm with the same laser power for each cell, and

the fluorescence emission was detected using a hybrid detector in photon counting mode at 498–551 nm. The data were analysed using a nandb—an R package for performing N&B analysis (Nolan et al, 2017).

## Total internal reflection microscopy

COS-7 cells grown on glass-bottom 35 mm Petri dishes (Mattek) were transiently co-transfected with PlexA-mClover, wild-type Sema1b-mRuby2 and all three Sema1b mutants. The same constructs were used for FRET-FLIM or Number and Brightness analysis. Two days post-transfection, the cells were washed with PBS and Dulbecco's modified eagle medium was replaced with a phenol red-free Dulbecco's modified eagle medium equilibrated at 37°C. For mouse Sema6A and PlxnA2, we used the same stable COS-7 cell lines as we used for FRET-FLIM. The images were acquired by a Zeiss Elyra TIRF microscope equipped with a 100× oil objective (1.46 NA). mRuby2 was excited at 561 nm, and the images were acquired at a resolution of 512 × 512 pixels (image size 49.7 × 49.7 μm). The average fluorescence intensity was calculated using ImageJ (Schindelin et al, 2012). In particular, TIRF micrographs were background subtracted to get rid of the EM-CCD camera noise. After this, each cell was profiled utilising a mask that only contained the signal coming from each cell and non-attributed-numbers for the background. The average grey value per each profiled cell was obtained and plotted as a box plot for each condition.

## Collapse assay of COS-7 cells

COS-7 cells grown on μ-Slides (chambered coverslips) with 8 wells (Ibidi) were transiently transfected with Drosophila melanogaster full-length PlexA-mClover (residues 28–1,945), wild-type Sema1b-mRuby2 (residues 37–686) or Sema1b-mutA-mRuby2 (residues 37–686) or Sema1b-mutB-mRuby2 (residues 37–686) or Sema1b-mutA+B-mRuby2 (residues 37–686). Two days post-transfection, the cells were washed with PBS and Dulbecco's modified eagle medium was replaced with a phenol red-free Dulbecco's modified eagle medium equilibrated at 37°C.

Images were acquired using a Leica SP8-X-SMD confocal microscope (Leica Microsystems) with a 63 × /1.40 numerical aperture oil immersion objective at 37°C. mClover and mRuby2 were excited at 488 and 561 nm, respectively, and the fluorescence emission was detected using two hybrid detectors in photon counting mode at 498–551 and 573–625 nm, respectively. Tiled positions (3 × 3) were scanned in 512 × 512 format every minute for 30 min. The pinhole was set at 3.0 Airy units, and an automatic adaptive autofocus was used to prevent z-drifting while imaging. After 2 min of imaging, a recombinant Sema1a-Fc was added to a final concentration of 5.8 μM. Cell surface area was calculated using ImageJ (Schindelin et al, 2012) before and after stimulation with recombinant Sema1a-Fc.

## DRG cultures

Culture methods were as previously described (Van Battum et al, 2014). In short, DRG neurons were dissected from E12.5 Sema6A knockout mouse embryos (Lilley et al, 2019). DRGs were collected in 1× Krebs medium (0.7% NaCl, 0.04% KCl, 0.02% KH$_2$PO$_4$, 0.2%

NaHCO$_3$ and 0.25% glucose) and dissociated by incubation with 0.25% trypsin in Krebs/EDTA for 10 min at 37°C. The reaction was halted by adding 2 mg soybean trypsin inhibitor, followed by trituration with a fire-polished Pasteur pipette in Krebs medium containing soybean trypsin inhibitor and 20 μg/ml DNAseI. Dissociated cells were resuspended in neurobasal medium supplemented with B-27, L-glutamine, penicillin/streptomycin, β-mercaptoethanol and nerve growth factor 2.5S (50 ng/ml, Alomone labs). Cells were plated onto poly-D-lysine (20 μg/ml) and laminin (10 μg/ml) coated glass coverslips in 12 wells plates in a humidified incubator at 37°C and 5% CO$_2$. Cultures were fixed by adding equal volume of 8% PFA in PBS containing 30% sucrose for 10–30 min at room temperature.

## Growth cone collapse assay

For growth cone collapse assays, DRG neurons were transfected at 1 day in vitro (DIV1) with Sema6A-WT-EGFP-pCAG, Sema6A-mutA-EGFP-pCAG, Sema6A-mutB-EGFP-pCAG or Sema6A-mutA+B-EGFP-pCAG mutant constructs or empty EGFP-pCAG vector using lipofectamine 2000 for 45 min at 37°C. At DIV4, vehicle (medium) or purified Sema6A-Fc was added to the cultures at a concentration of 1, 10 or 75 nM for 1 h at 37°C. Cultures were fixed and processed for immunocytochemistry with anti-GFP antibodies and counterstained with phalloidin to visualise F-actin in filopodia and lamellipodia to determine growth cone morphology. The following antibodies were used in this experiment: Rabbit anti GFP (Thermo Fisher Scientific, catalog # A-11122), Donkey anti-Rabbit IgG (H+L) Alexa Fluor 488 (Thermo Fisher Scientific, catalog # A-21206) and Alexa Fluor 568 Phalloidin (Invitrogen, catalog # A-12380). Coverslips were mounted and scored for growth cone morphology using a scale from one to four (uncollapsed) and five to eight (fully collapsed) according to a matrix of growth cones with different morphologies, allowing the detection of even subtle changes in growth cone morphology (van Erp et al, 2015; Kong et al, 2016). Assigning each growth cone to its appropriate category was based on the following criteria: Categories 1-4 show uncollapsed growth cones, with category 1 showing multiple extending filopodia that are reduced in number in category 2. Category 3 and 4 growth cones both lack filopodia with category 4 growth cones additionally showing F-actin reduction. Categories 5–8 show collapsed growth cones, with growth cone size and shape reducing to no discernible growth cone at all in category 8. Images were acquired on an epifluorescence miscroscope (Zeiss Axioscope A1). A categorical analysis using Fisher's exact test was used to test for statistical significance of uncollapsed or fully collapsed growth cones for each mutant or control construct. Statistical tests were performed using IBM SPSS Statistics 23.

## Animals

All animal use and care was in accordance with institutional guidelines and approved by the animal experimentation committee (DEC). The mouse strain was Sema6A (Sema6aGt[KST069]Byg) kept on a C57/B6J background. Timed-pregnant females were 3–6 months of age. Timed-pregnant mice were euthanised by means of cervical dislocation. The morning on which a vaginal plug was detected was considered embryonic day 0.5 (E0.5).

## Data availability

Structure factors and coordinates have been deposited in the Protein Data Bank with identification numbers PDB: 6FKM (https://www.rcsb.org/structure/6FKM) and 6FKN (https://www.rcsb.org/structure/6FKN).

**Expanded View** for this article is available online.

## Acknowledgements
We thank the staff of Diamond Light Source for support and access to beamlines I03 and I24; Weixian Lu for help with tissue culture; Thomas Walter for crystallisation technical support; David Staunton for assistance with biophysical experiments. The work was funded by Cancer Research UK and Medical Research Council Programme Grants (to E.Y.J., C375/A17721 and MR/M000141/1) and the Netherlands Organization for Scientific Research (to R.J.P., ALW-VICI). The Wellcome Centre for Human Genetics is supported by Wellcome Trust Centre grant 203141/Z/16/Z. D.R. was supported by EMBO Long-Term Fellowship (ALTF 604-2014) and S. P. P. by the Nuffield Department of Medicine Leadership Fellowship.

## Author contributions
Conceptualisation, DR, RJP and EYJ; Methodology, DR, RAR, LA, SP-P and EYJ; Investigation, DR, MGV, DK, GNN, LA and KH, Writing, DR and EYJ, Funding acquisition, DR, SP-P, RJP and EYJ; Supervision, SP-P, RJP and EYJ.

## Conflict of interest
The authors declare that they have no conflict of interest.

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
