## [Review Process File · The EMBO Journal]

Structural basis of semaphorin-plexin cis interaction

E. Yvonne Jones, Daniel Rozbesky, Marieke Verhagen, Dimple Karia, Gergely Nagy, Luis Alvarez, Ross Robinson, Karl Harlos, Sergi Padilla-Parra, and R. Pasterkamp

DOI: 10.15252/embj.2019102926

Corresponding author(s): E. Yvonne Jones (yvonne@strubi.ox.ac.uk) and R. Pasterkamp r.j.pasterkamp@umcutrecht.nl

Review Timeline:

Submission Date:	11th Jul 19
Editorial Decision:	19th Aug 19
Revision Received:	16th Nov 19
Editorial Decision:	12th Dec 19
Revision Received:	18th Mar 20
Editorial Decision:	9th Apr 20
Revision Received:	22nd Apr 20
Accepted:	23rd Apr 20

Editor: Karin Dumstrei

Transaction Report:

Dear Yvonne,

Thank you for submitting your manuscript to The EMBO Journal. Your study has now been seen by three referees and their comments are provided below.

As you can see from these comments, the referees find the analysis interesting and insightful. They raise a number of constructive comments that I would like to ask you to address in a revised version. In particular some further analysis is needed to support the "cis" interaction between semaphorin-plexin. The concerns raised are sensible and I anticipate that you will be able to resolve them in a good way. Let me know if we need to discuss any of the points in more detail - happy to do so.

Thank you for the opportunity to consider your work for publication. I look forward to your revision.

with best wishes

Karin

Karin Dumstrei, PhD
Senior Editor
The EMBO Journal

When assembling figures, please refer to our figure preparation guideline in order to ensure proper formatting and readability in print as well as on screen:
<http://bit.ly/EMBOPressFigurePreparationGuideline>

- a point-by-point response to the referees' comments, with a detailed description of the changes made (as a word file).

- a word file of the manuscript text.

- individual production quality figure files (one file per figure)

- a complete author checklist, which you can download from our author guidelines

(<https://www.embopress.org/page/journal/14602075/authorguide>).

- Expanded View files (replacing Supplementary Information)

Further information is available in our Guide For Authors:

The revision must be submitted online within 90 days; please click on the link below to submit the revision online before 17th Nov 2019.

Link Not Available

Referee #1:

Semaphorins activate downstream signaling by directly or indirectly binding to a member of the plexin family of receptors. Previous studies showed that some membrane-bound semaphorins interact with plexins also in cis when expressed in the same cell. Rozbesky et al. now report two crystal structures for complexes between *Drosophila* Sema1b and PlexA that reveal a binding site for the cis interaction. The cis interaction is mediated by two binding sites (site A and site B) in the Sema1b semaphorin domain. Site A mediates a head-to-head binding while site B is a novel binding surface for an interaction mode described as side-on interaction. This site is specific for monomeric Sema1b because it is occluded in dimerized Sema1b. The authors show that mouse Sema6A employs a similar binding mode for its cis interaction with mammalian Plexin-A2.

To determine the role of the two binding sites the authors created mutations in site A and B and analyzed their effect on the interaction with plexins. Plexins assume an auto-inhibited, ring-like conformation in the absence of ligand. Single particle EM and FLIM-FRET measurements indicate a conformational change with an opening of the ring upon binding of Sema1b or Sema6A but it remains to be shown that this affects ligand binding or receptor function. Sema1b and Sema6A constructs with a mutation in site A or B still induce a conformational change in the plexin ectodomain but double mutants no longer interact with plexins. To determine the functional significance of the binding sites the authors performed collapse assays using COS7 cells as a

heterologous system for Sema1b and PlexA and sensory neurons from mouse dorsal root ganglia for Sema6A. Co-expression of *Drosophila* Sema1b and mammalian Sema6A inhibits the collapse induced by the activation of PlexA and Plexin-A2, respectively. Only mutation of both site A and B blocks the inhibitory effect on receptor activation. However, the conclusions that can be drawn from the collapse assays are limited because only a single ligand concentration is used. It has been shown that Sema6A can induce the collapse of sensory growth cones at high concentrations (Xu et al. 2000), which were not tested.

The structural analysis by Rozbesky et al. makes an important contribution by providing the first structure for a cis interaction, which will help to better understand plexin regulation. The authors suggest that the cis interaction modulates the threshold for receptor activation. They propose that binding via site A interferes with ligand binding while interaction through site B prevents receptor activation by sterically blocking the engagement of a second plexin to form an active complex. However, additional functional analyses are required to confirm this model

Specific points:

- To test their functional significance point mutations were introduced into site A and B of Sema1b and Sema6A. The mutations in Sema1b and especially the introduction of glycosylation sites into Sema6A could affect the intracellular trafficking and/or endocytosis of the semaphorins or the activated semaphorin/plexin complexes but it was not verified that they are expressed at comparable levels or show the same subcellular localization and cell surface expression in COS7 cells and sensory neurons.

- For the collapse assays, only a single ligand concentration was used. A dose-response analysis and a calculation of the EC50 are necessary to characterize the effect of mutations in site A or B. In addition, the authors should determine how the affinity of the plexin/semaphorin complexes for their ligands is affected by mutations in site A or B. Mutation of site B increases its affinity to PlexA compared to wild type Sema1b (Fig. 1E). The mutations may differentially affect also other properties of Sema1b and Sema6A.

- A scale from 1 to 8 is used to score the collapse of growth cones from sensory neurons. The authors used the same scale in a previous publication (Kong et al., 2016) but did not explain how these categories are defined. The difference between e.g. score 2 - 4 or 5 and 6 is not apparent from the examples shown in Fig. 3. The figure legend mentions a statistical test but no standard error or significance is indicated in the figure. The number of independent experiments is not indicated. The effect of incubation with vehicle only (medium without semaphorin) has to be quantified for all combination of constructs as an essential control for the collapse assay (Fig. 2F and 3A).

- In a separate study, Rozbesky et al. show that *Drosophila* Sema1b is a monomer in solution. To demonstrate that monomers are present also on the cell surface the authors rely exclusively on a method called number and brightness analysis. The basis for this method is not explained in the manuscript and it is not shown that it can be applied to the dimerization of single molecules. Digman et al. (2008) developed this analysis to detect and quantify changes in molecular aggregates. They tested this method by following the disassembly of relatively large paxillin complexes. Rozbesky et al. do not provide evidence that this method can be used for the formation of dimers. It also cannot be excluded that the changes in the brightness values shown in Fig. S1 result from effects other than dimerization, e.g. the recruitment to specific membrane domains or a

concentration in vesicles after endocytosis. Fig. S1 also suggests that monomeric and artificially dimerized Sema1b show a different subcellular distribution. Are the experimental conditions in Fig. S1D comparable to those in Fig. 3? Sema1b induces a collapse in Fig. 3G-K but not in Fig. S1D.

- The figure legends do not include sufficient information. While image acquisition is explained in detail in Material and Methods insufficient information is given for the collapse assay (e.g. incubation time with ligand).

Referee #2:

This paper by Rozbesky et al reports two crystal structures of the drosophila Sema1b-PlexA complex. In one of the structures, a novel interface (termed "side-on") between Sema1b and PlexA is formed, in addition to the canonical semaphorin-plexin head-to-head interface. Unlike other semaphorin, Sema1b is unique in that it does not dimerize, and therefore the head-to-head interaction does not lead to the dimerization of PlexA that is required for its activation. Biophysical experiments suggest that the side-on interaction does occur in solution. Cell-based experiments suggest that formation of the side-on interaction between Sema1b and Plexin, and between mammalian Sema6A and Plexin, plays an inhibitory role in plexin signaling by preventing the dimerization of plexin induced by the dimeric semaphorin in trans. These experiments led to a model that potentially underlies the inhibitory regulation of plexin by cis-acting semaphorin, which has been reported by many previous studies but without a clear mechanistic explanation. The proposed model of the paper is novel and interesting, and potentially provides an unanticipated structural basis for an important aspect of plexin regulation. The paper is clearly written.

One major concern I have is on the nature of the novel side-on interface. As pointed out in the paper, the loop between residues 353 and 373 in Sema1b (Exb1-b2 loop), while involved in the side-on interface, is disordered in the 2:2 structure. In the 1:1 complex structure, however, this loop is well resolved, although it is not involved in any packing interaction. These observations are counterintuitive as one would expect that the loop should be better structured when it is engaged in specific interactions. Moreover, it is not obvious how this loop could be arranged to avoid clash with PlexA because this part of Sema1b is deeply imbedded in the interface and there is little space for the loop to pass in the crystal lattice. Is it possible that this loop is cleaved somewhere in the middle during purification or crystallization in the 2:2 complex? This is an important point because the paper hinges on the interactions involving this loop, and the mutations for testing the side-on interface are targeted to this loop.

The structural data suggest that the side-on interface between Sema1b and PlexA is likely much weaker than the canonical head-to-head interface, because it is not present in the 1:1 structure, and in the 2:2 structure only one such interface is formed. Also, the side-on interface is much smaller than the head-to-head interface. In contrast, the MST data show that the affinity of the side-on interface is similar to that of the head-to-head interface, with both in the nanomolar range. In fact, the side-on interface seems to dominate in other experiments. One possible reason for this discrepancy might be that the protein used in crystallization was de-glycosylated, which weakens the interaction mediated by the glycosylation group on Asn289. Related to this point, based on the structure, both interfaces can simultaneously form in solution. What is the stoichiometry of the Sema1b/PlexA complex in solution measured by light scattering and/or AUC?

The MST titration data for the sema1b WT are missing in Figure S2. In addition, the middle row of

Figure S2 is difficult to see. The changes in fluorescence appear to be similar in E, H and K, which seems to contradict with the dramatically different data and fit in F, I and L, respectively. Which part of the experimental traces was used for fitting in the right column? A panel with expanded views of the part of the traces used for analyses could help clarify this. Have the authors considered using a different technique, such as SPR, to measure the binding and confirm the binding affinities?

The choice of mutations for testing the side-on interface is a bit puzzling to me. Based on the structure, point mutations should be easy to design to test the three parts of the interface, instead of alanine mutations of the entire Exb1-b2 loop. Strong effects of mutating the glycosylation site Asn289 could be of great support of the model.

Minor points:

The results in Figure 2A suggests that MutA+B maintains significant binding to PlexA? This is odd because this mutant contains so many mutations and has been shown by MST to completely abolish binding to PlexA.

Page 7, second and fourth paragraphs: References to figures should be Figure 1, instead of Figure 2.

Referee #3:

In this most interesting study, the authors observed a new "cis" interaction between Drosophila semaphoring Sema1b and mouse PlexA-Sema6A. Previously, the authors discovered that Sema1b forms monomers, whereas all other semaphorins with known structure form disulfide-stabilized dimers. Here, the authors obtained a crystal form of the Sem1b / PlexA-Sema6A complex where two distinct interfaces form. One is an interface that has been previously observed, while the other, "cis" interface is enabled by the homomeric nature of Sema1b. FRET experiments suggest that Sema1b opens the ring that is formed by the other semaphorins when bound to PlexA. Structure-based mutations of both interfaces of Sema1b were tested by binding experiments, FRET experiments that probe PlexA ring formation, and cone morphology.

I recommend publication after revision, as outlined below.

1. Please provide more molecular details about both interfaces, including buried interface areas, and types of interactions (e.g., specific sidechain interactions, charge complementarity, shape fitting). Also, how did the authors decide on the choices of A and B mutations?

2. It is implied that the B (or "cis") interface would be incompatible with the homodimerization interface that is observed in the other semaphorins. Please provide a superposition of the Sema1b-Sema6A cis-interface with a representative homodimer of one of the other semaphorins.

3. The effects of the B mutant on FRET are rather modest in Figures 2E and S4, certainly not as large as suggested in the cartoon in Figure 2D. Thus, the statements "Binding of Sema1b to PlexA in cis leads to opening ..." (similarly for the Sema6 system) should be softened considerably.

4. Did the authors attempt negative stain EM studies of the rings produced by PlexA in the

presence of Sema1b and its A/B mutants?

5. Considering that the effects of the Sema1b A, B, A+B mutations on PlexA ring formation (as assessed by FRET) are rather small (Figure 2A), but that the effect on growth cone morphology is much larger upon mutation of both A and B interfaces, please speculate about possible models that might explain these somewhat divergent observations.

Point-by-point response to the reviewers' comments

Manuscript: Manuscript EMBOJ-2019-102926

Title: Structural basis of semaphorin-plexin *cis* interaction

Authors: Daniel Rozbesky, Marieke G. Verhagen, Dimple Karia, Gergely N. Nagy, Luis Alvarez, Ross A. Robinson, Karl Harlos, Sergi Padilla-Parra, R. Jeroen Pasterkamp, and E. Yvonne Jones

November 15, 2019

We thank the reviewers for their interest in our work and constructive comments.

Referee #1:

Semaphorins activate downstream signaling by directly or indirectly binding to a member of the plexin family of receptors. Previous studies showed that some membrane-bound semaphorins interact with plexins also in *cis* when expressed in the same cell. Rozbesky et al. now report two crystal structures for complexes between *Drosophila* Sema1b and PlexA that reveal a binding site for the *cis* interaction. The *cis* interaction is mediated by two binding sites (site A and site B) in the Sema1b semaphorin domain. Site A mediates a head-to-head binding while site B is a novel binding surface for an interaction mode described as side-on interaction. This site is specific for monomeric Sema1b because it is occluded in dimerized Sema1b. The authors show that mouse Sema6A employs a similar binding mode for its *cis* interaction with mammalian Plexin-A2.

To determine the role of the two binding sites the authors created mutations in site A and B and analyzed their effect on the interaction with plexins. Plexins assume an auto-inhibited, ring-like conformation in the absence of ligand. Single particle EM and FLIM-FRET measurements indicate a conformational change with an opening of the ring upon binding of Sema1b or Sema6A but it remains to be shown that this affects ligand binding or receptor function. Sema1b and Sema6A constructs with a mutation in site A or B still induce a conformational change in the plexin ectodomain but double mutants no longer interact with plexins. To determine the functional significance of the binding sites the authors performed collapse assays using COS7 cells as a heterologous system for Sema1b and PlexA and sensory neurons from mouse dorsal root ganglia for Sema6A. Co-expression of *Drosophila* Sema1b and mammalian Sema6A inhibits the collapse induced by the activation of PlexA and Plexin-A2, respectively. Only mutation of both site A and B blocks the inhibitory effect on receptor activation. However, the conclusions that can be drawn from the collapse assays are limited because only a single ligand concentration is used. It has been shown that Sema6A can induce the collapse of sensory growth cones at high concentrations (Xu et al. 2000), which were not tested.

Although the study by Xu et al. (2000) shows Sema6A-mediated growth cone collapse of chick DRG neurons (at 1 nM), work by Haklai-Topper et al. (2010) does not show collapse of mouse DRG growth cones by Sema6A at similar concentrations. Higher concentrations can indeed cause collapse but even at these concentrations Sema6A KO neurons are much more sensitive to Sema6A application.

The structural analysis by Rozbesky et al. makes an important contribution by providing the first structure for a *cis* interaction, which will help to better understand plexin regulation. The authors suggest that the *cis* interaction modulates the threshold for receptor activation. They propose that binding via site A interferes with ligand binding while interaction through site B prevents receptor activation by sterically blocking the engagement of a second plexin to form an active complex.

However, additional functional analyses are required to confirm this model

Specific points:

- To test their functional significance point mutations were introduced into site A and B of Sema1b and Sema6A. The mutations in Sema1b and especially the introduction of glycosylation sites into Sema6A could affect the intracellular trafficking and/or endocytosis of the semaphorins or the activated semaphorin/plexin complexes but it was not verified that they are expressed at comparable levels or show the same subcellular localization and cell surface expression in COS7 cells and sensory neurons.

We thank the reviewer for raising this point. We agree that the representative images showing the comparable expression levels were not shown for FRET-FLIM analysis of Sema6A-PlxnA2 and for collapse assays. To demonstrate the similar expression level and localization on the cell surface, we performed a semi-quantitative analysis in single cells to determine the level of intensity for each condition. We have now included the representative images showing the comparable expression levels on the cell surface for FRET-FLIM analysis of Sema6A-PlxnA2 (Fig. EV4B) for COS-7 collapse assay (Fig. EV3G-K) and for growth cone collapse assay in DRG neurons (Fig. 3).

- For the collapse assays, only a single ligand concentration was used. A dose-response analysis and a calculation of the EC50 are necessary to characterize the effect of mutations in site A or B.

For the collapse assays, we respectfully submit that the dose-response analysis is not feasible. For EC50 calculation, we would need to generate the dose-response curve for three mutants (A, B, A+B) and one wild-type protein of each Sema1b and Sema6A. Each dose-response curve should contain at least 15 different concentrations of a particular semaphorin that would be set up in triplicates. That would require measurement and analysis of at least 360 different conditions (4x2x15x3). In practice, the calculation of an EC50 just for the Sema1b-PlexA system in COS-7 cells would take at least 360 hours at the confocal microscope and tens of milligrams of purified semaphorin (depending on Emax). Several previous publications, including our own, show that Sema6A can cause growth cone collapse in for example sympathetic and dentate gyrus neurons at around 1 nM (Xu et al., 2000; Van Battum et al., 2017). In the current study we use 75 nM, a concentration at which a clear differences in the DRG growth cone collapse response has been reported between WT and Sema6A KO neurons (Haklai-Topper et al., 2010). This difference forms the basis of the experiments outlined in this study.

In addition, the authors should determine how the affinity of the plexin/semaphorin complexes for their ligands is affected by mutations in site A or B. Mutation of site B increases its affinity to PlexA compared to wild type Sema1b (Fig. 1E).

As also discussed in our response to Reviewer 2, we initially set out to use standard methods to fully characterize the binding affinity and kinetics of the wild type Sema1b – PlexA interaction. However, we were not able to use surface plasmon resonance (SPR) binding assays because Sema1b and PlexA molecules showed high non-specific binding onto the surface of CM5 or SA sensor chips. Also, we were not able to use ITC (isothermal titration calorimetry) measurements because *Drosophila* Sema1b and PlexA were not expressed in quantities sufficient for ITC. The Sema1b mutants were produced as tools to probe the functionality of site A and site B binding and although we agree it would be interesting to use them to dissect the biophysical properties of these two binding surfaces, we respectfully submit that this is beyond the scope of the current study.

The mutations may differentially affect also other properties of Sema1b and Sema6A.

Regarding the mutation effect on protein structure, all Sema1b and Sema6A mutants were expressed and secreted at similar levels to the wild-type protein in HEK293T cells. Furthermore, wild-type and all mutants were eluted at the same time from the size-exclusion column suggesting there is no problem with folding. This information has been added to the text to address the reviewer's concern (page 8).

- A scale from 1 to 8 is used to score the collapse of growth cones from sensory neurons. The authors used the same scale in a previous publication (Kong et al., 2016) but did not explain how these categories are defined. The difference between e.g. score 2 - 4 or 5 and 6 is not apparent from the examples shown in Fig. 3. The figure legend mentions a statistical test but no standard error or significance is indicated in the figure. The number of independent experiments is not indicated. The effect of incubation with vehicle only (medium without semaphorin) has to be quantified for all combination of constructs as an essential control for the collapse assay (Fig. 2F and 3A).

We thank the reviewer for these comments. As the reviewer indicates we have used a similar matrix in several of our previous studies, including Kong et al. (2016) and Van Erp et al. (2015). Growth cone collapse was graded using a scale from one (uncollapsed) to eight (fully collapsed) according to a matrix of growth cones with different morphologies, allowing the detection of even subtle changes in growth cone morphology. These growth cones were selected from the current dataset as growth cone morphologies can differ between cell types (for selection of morphologies, please see Figure S6 from Van Erp et al, 2015). To address the reviewer's concern we have improved the current matrix by including growth cones with more distinct morphologies. Further, we have included a reference to the study by Van Erp et al (2015). We would also like to note that statistics was not performed by comparing the 8 different groups of growth cones, but by comparing groups 1-4 with 5-8. Therefore, no SD or SEM is shown. We have improved the description of this statistical analysis in the Methods sections. Further, the number of independent experiments is now indicated in the legend. Finally, we respectfully disagree that our analysis would even be stronger by addition of medium controls for every group. Our results in Figure 3 clearly show growth cone collapse induced by Sema6A in Sema6A KO neurons. This is line with previous work (Haklai-Topper et al., 2010) and the basis of the rescue experiments that we perform. Indeed, re-introduction of full-length Sema6A in Sema6A KO neurons dramatically reduces Sema6A-induced growth cone collapse.

We have investigated the effect of incubation with vehicle only for COS-7 cells, please see Fig. EV3 L.

- In a separate study, Rozbesky et al. show that *Drosophila* Sema1b is a monomer in solution. To demonstrate that monomers are present also on the cell surface the authors rely exclusively on a method called number and brightness analysis. The basis for this method is not explained in the manuscript and it is not shown that it can be applied to the dimerization of single molecules.

We thank the reviewer for this remark. We have now introduced the basics of this method in the main text (Page 6). We have also included references from our own work in which we clearly show how Number and Brightness can detect and quantify the transition between monomers to dimers (Nolan et al., 2017 Bioinformatics, Nolan et al., 2018, Methods, Nolan et al., 2018 Jove and Iliopoulou et al., 2018 NSMB).

Digman et al. (2008) developed this analysis to detect and quantify changes in molecular aggregates. They tested this method by following the disassembly of relatively large paxillin complexes. Rozbesky et al. do not provide evidence that this method can be used for the formation of dimers.

We thank the reviewer for citing the seminal work of Enrico Gratton and their application. We have developed further the method by implementing a novel detrending algorithm that can detect monomers and dimers both in live cells (Nolan et al., 2017 *Bioinformatics*, Iliopoulou et al., 2018 *NSMB*) and in vitro (Nolan et al., 2018 *JOVE*). We have added this information in the main text (Page 6).

It also cannot be excluded that the changes in the brightness values shown in Fig. S1 result from effects other than dimerization, e.g. the recruitment to specific membrane domains or a concentration in vesicles after endocytosis. Fig. S1 also suggests that monomeric and artificially dimerized Sema1b show a different subcellular distribution.

For measurement of Number and Brightness, we acquired images for individual cells with comparable intensities. We have not observed any changes in trafficking or subcellular localization. Also, the confocal images were taken from the membrane, thus potential changes in trafficking should not affect the monomeric or dimeric state on the cell surface. To demonstrate this point, we already provided the images (pixel-by-pixel analysis) for both the intensity and the brightness in our initial manuscript (Fig. S1B, D; now Fig. EV1B, EV1D).

Are the experimental conditions in Fig. S1D comparable to those in Fig. 3? Sema1b induces a collapse in Fig. 3G-K but not in Fig. S1D.

Thank you for this comment about Fig. S1D (now Fig EV1D) versus Fig 3. We used different constructs of PlexA. We used PlexA_{FL}-mClover (the full-length PlexA fused with fluorescent protein mClover) for the collapse assay while PlexA-mClover (the ectodomain followed by a transmembrane segment and the C-terminal fluorescent protein mClover) was used for the Number and Brightness analysis and FRET-FLIM. Details of the constructs are presented in Materials and Methods; however, that they are different is an important point and we have now highlighted this in the main text (page 6), Methods (page 21) and figure legends S1 (now EV1).

- The figure legends do not include sufficient information. While image acquisition is explained in detail in Material and Methods insufficient information is given for the collapse assay (e.g. incubation time with ligand).

Thank you, we have provided more information for the collapse assay – figure legend 2, 3 and S3 (now EV3).

Referee #2:

This paper by Rozbesky et al reports two crystal structures of the drosophila Sema1b-PlexA complex. In one of the structures, a novel interface (termed "side-on") between Sema1b and PlexA is formed, in addition to the canonical semaphorin-plexin head-to-head interface. Unlike other semaphorin, Sema1b is unique in that it does not dimerize, and therefore the head-to-head interaction does not lead to the dimerization of PlexA that is required for its activation. Biophysical experiments suggest that the side-on interaction does occur in solution. Cell-based experiments suggest that formation of the side-on interaction between Sema1b and Plexin, and between mammalian Sema6A and Plexin, plays an inhibitory role in plexin signaling by preventing the dimerization of plexin induced by the

dimeric semaphorin in trans. These experiments led to a model that potentially underlies the inhibitory regulation of plexin by cis-acting semaphorin, which has been reported by many previous studies but without a clear mechanistic explanation. The proposed model of the paper is novel and interesting, and potentially provides an unanticipated structural basis for an important aspect of plexin regulation. The paper is clearly written.

One major concern I have is on the nature of the novel side-on interface. As pointed out in the paper, the loop between residues 353 and 373 in Sema1b (Exb1-b2 loop), while involved in the side-on interface, is disordered in the 2:2 structure. In the 1:1 complex structure, however, this loop is well resolved, although it is not involved in any packing interaction. These observations are counterintuitive as one would expect that the loop should be better structured when it is engaged in specific interactions. Moreover, it is not obvious how this loop could be arranged to avoid clash with PlexA because this part of Sema1b is deeply imbedded in the interface and there is little space for the loop to pass in the crystal lattice. Is it possible that this loop is cleaved somewhere in the middle during purification or crystallization in the 2:2 complex? This is an important point because the paper hinges on the interactions involving this loop, and the mutations for testing the side-on interface are targeted to this loop.

We thank the reviewer for raising this important point. We agree that the Ex β 1- β 2 loop is well resolved in the 1:1 complex; however, we do not think that the loop is disordered in the 2:2 complex. Unfortunately, the low resolution of the 2:2 complex (4.8 Å) and the poor and fragmentary electron density in this region does not allow us to build the loop unambiguously, and this is why we have not included it in our model. The Ex β 1- β 2 loop has to adopt a different orientation in order to avoid steric clashes with PlexA₁₋₄ in the 2:2 complex and although we cannot define specific interactions because of the quality of the electron density map, the loop is positioned such as to be able to contribute interactions to stabilize the complex. We have clarified this point in the main text – page 7-8.

We do not think that the loop was cleaved during the production, purification or crystallization. First, Sema1b₁₋₂ was expressed in HEK293T cells and secreted to the media. HEK293T cells usually do not express a large number of proteases that could lead to the specific cleavage of the Ex β 1- β 2 loop. We have in-hand experience with unwanted cleavage by furin protease in HEK293T cells; however, there is no furin cleavage site in the Ex β 1- β 2 loop. Second, crystals of the 2:2 complex showed up much faster (15 days) than the crystals of the 1:1 complex (132 days). If the loop had been cleaved during purification or crystallization, it would have been likely also cleaved in the 1:1 complex. Third, as discussed above, we can see the electron density fragments for the loop; however, we are not able to unambiguously fit the loop in the electron density because of the low resolution.

The structural data suggest that the side-on interface between Sema1b and PlexA is likely much weaker than the canonical head-to-head interface, because it is not present in the 1:1 structure, and in the 2:2 structure only one such interface is formed. Also, the side-on interface is much smaller than the head-to-head interface. In contrast, the MST data show that the affinity of the side-on interface is similar to that of the head-to-head interface, with both in the nanomolar range. In fact, the side-on interface seems to dominate in other experiments. One possible reason for this discrepancy might be that the protein used in crystallization was de-glycosylated, which weakens the interaction mediated by the glycosylation group on Asn289. Related to this point, based on the structure, both interfaces can simultaneously form in solution. What is the stoichiometry of the Sema1b/PlexA complex in solution measured by light scattering and/or AUC?

We again thank the reviewer for this elegant explanation. We have now commented on this possibility in the revised manuscript (page 9).

We have performed additional SEC-MALS measurements (new Fig. EV2 M-O) to address the reviewer's question about the stoichiometry of the Sema1b-PlexA complex in solution. SEC-MALS revealed three peaks corresponding to the unliganded Sema1b, unliganded PlexA and the Sema1b in complex with PlexA in 1:1 stoichiometry. Interestingly the majority of the proteins are in the monomeric state, presumably the kinetics of the site A and site B interactions in vitro are such that only an undetectably small proportion of the molecules engage in a tetrameric complex. However, as further investigation of the biophysics of the system in solution is beyond the scope of this manuscript we have included the data without speculation.

The MST titration data for the sema1b WT are missing in Figure S2. In addition, the middle row of Figure S2 is difficult to see. The changes in fluorescence appear to be similar in E, H and K, which seems to contradict with the dramatically different data and fit in F, I and L, respectively. Which part of the experimental traces was used for fitting in the right column? A panel with expanded views of the part of the traces used for analyses could help clarify this. Have the authors considered using a different technique, such as SPR, to measure the binding and confirm the binding affinities?

We apologise and are very grateful to the reviewer for spotting this omission. The wild-type and mutant data were collected at the same time, however, we included the wild-type data in a comparison of the binding affinities of all fly semaphorin class 1/2 and PlexA/B pairs that formed part of our recently published study on fly semaphorin structures (Rozbesky et al. Nat Commun 2019). We have now clarified this point in the Figure legend 1 to reference the source of the wild-type curve. We also explicitly note that the wild-type and mutant data were collected at the same time (in triplicate).

We agree with the reviewer that the changes in fluorescence appear to be similar in E, H and K (Figure S2 – now Fig EV2). We have now added the hot and cold regions that were used for fitting as well as close-up views for the hot regions to demonstrate differences in binding properties. We have also clarified this point in the Figure legend EV2.

Regarding SPR, initially, we wanted to use standard methods for the analysis of protein-protein interactions; however, we were not able to use surface plasmon resonance (SPR) binding assays because Sema1b and PlexA molecules showed high non-specific binding onto the surface of CM5 or SA sensor chips. Also, we were not able to use ITC (isothermal titration calorimetry) measurements because *Drosophila* Sema1b and PlexA were not expressed in quantities sufficient for ITC.

The choice of mutations for testing the side-on interface is a bit puzzling to me. Based on the structure, point mutations should be easy to design to test the three parts of the interface, instead of alanine mutations of the entire Exb1-b2 loop. Strong effects of mutating the glycosylation site Asn289 could be of great support of the model.

Given the low resolution of the 2:2 complex, and consequent lack of detailed information on residue-to-residue interactions, we decided to test the side-on interface by replacing each residue in the Exβ1-β2 loop by alanine rather than simple point mutations. We have clarified this point in page 8 of the revised manuscript. We chose the Exβ1-β2 loop for mutagenesis because, as discussed earlier, the loop potentially plays an important role during the complex formation, and it provided an attractive candidate in which to introduce a substantial mutation. As this mutation did indeed provide us with the required reagent to test the function of site B binding we did not pursue further mutagenesis-based mapping of the binding site. We agree with the reviewer that such mapping, including an analysis of the contribution of the glycosylation, is of interest, but respectfully argue that this analysis is beyond the scope and aims of the current study.

Minor points:

The results in Figure 2A suggests that MutA+B maintains significant binding to PlexA? This is odd because this mutant contains so many mutations and has been shown by MST to completely abolish binding to PlexA.

We agree that the results in Fig. 2A suggest that Sema1b-mutA+B maintains binding to PlexA on the cell surface of COS-7 cells. However, the binding does not appear to be as strong as that observed for Sema1b wild-type or mutA or mutB. Indeed, the average lifetime for mutA+B is significantly higher than those lifetimes observed for wild-type, mutA and mutB, and is more similar to the lifetime of donor alone. It is possible that increased concentrations of PlexA and Sema1b on the cell surface resulted in non-specific binding. Furthermore, these findings are not in conflict with our MST data because we used only the first four domains of the PlexA ectodomain in MST whereas a PlexA construct encompassing the full-length ectodomain including transmembrane segment and a short part of cytoplasmic domain was used in FRET-FLIM.

Page 7, second and fourth paragraphs: References to figures should be Figure 1, instead of Figure 2.

Thank you, we have corrected Fig 2 to Fig 1 – Page 7.

Referee #3:

In this most interesting study, the authors observed a new "cis" interaction between *Drosophila* semaphoring Sema1b and mouse PlexA-Sema6A. Previously, the authors discovered that Sema1b forms monomers, whereas all other semaphorins with known structure form disulfide-stabilized dimers. Here, the authors obtained a crystal form of the Sema1b / PlexA-Sema6A complex where two distinct interfaces form. One is an interface that has been previously observed, while the other, "cis" interface is enabled by the homomeric nature of Sema1b. FRET experiments suggest that Sema1b opens the ring that is formed by the other semaphorins when bound to PlexA. Structure-based mutations of both interfaces of Sema1b were tested by binding experiments, FRET experiments that probe PlexA ring formation, and cone morphology.

I recommend publication after revision, as outlined below.

1. Please provide more molecular details about both interfaces, including buried interface areas, and types of interactions (e.g., specific sidechain interactions, charge complementarity, shape fitting). Also, how did the authors decide on the choices of A and B mutations?

Thank you for raising this point. For individual mutations in Sema1b-mutA, we selected sequences that were surface exposed and located at the interaction interface. As the interaction interface is relatively large, we decided to use three single point mutations, namely, F203E, Q219R and K223E. The single point mutations were designed to introduce electrostatic repulsions or reduction of surface hydrophobicity.

As discussed in our response to Reviewer 2, given the low resolution of the 2:2 complex, and consequent lack of detailed information on residue-to-residue interactions, we decided to test the side-on interface by replacing each residue in the Exβ1-β2 loop by alanine rather than simple point mutations. We have clarified this point in page 8 of the revised manuscript. We chose the Exβ1-β2 loop for mutagenesis because, as discussed earlier, the loop potentially plays an important role

during the complex formation, and it provided an attractive candidate in which to introduce a substantial mutation.

We agree that it would be interesting to provide more molecular details about both interfaces; however, we think that the resolution of the 2:2 complex (4.8 Å) is not sufficient to provide exact and unambiguous details. In particular, buried interface area, charge complementarity and shape fitting rely on precise conformations of side chains. Furthermore, we were not able to completely build the Exβ1-β2 loop of Sema1b, which is also involved in binding to PlexA, because of the fragmentary electron density in this region (please see our discussion of this point in response to reviewer 2). Thus, we respectfully argue that it is not possible to provide a detailed comparison of the characteristics of both interfaces.

2. It is implied that the B (or "cis") interface would be incompatible with the homodimerization interface that is observed in the other semaphorins. Please provide a superposition of the Sema1b-Sema6A cis-interface with a representative homodimer of one of the other semaphorins.

We thank the reviewer for this useful suggestion. We have now added Fig S1H (now Fig. EV1H) showing a superposition of the Sema1b – PlexA cis complex and the crystal structure of *Drosophila* Sema2a dimer (crystal structure from Rozbesky et al. Nat Commun 2019).

3. The effects of the B mutant on FRET are rather modest in Figures 2E and S4, certainly not as large as suggested in the cartoon in Figure 2D. Thus, the statements "Binding of Sema1b to PlexA in cis leads to opening ..." (similarly for the Sema6 system) should be softened considerably.

Thank you for raising this point. We have now modified the cartoon in Figure 2D and softened our conclusions (please see page-31).

4. Did the authors attempt negative stain EM studies of the rings produced by PlexA in the presence of Sema1b and its A/B mutants?

We made several attempts to characterize head-to-head and side-on complexes of Sema1b and PlexA using single particle negative stain EM. However, we did not observe intact complex particles because the complex fell apart, probably due to the low pH of uranyl formate.

5. Considering that the effects of the Sema1b A, B, A+B mutations on PlexA ring formation (as assessed by FRET) are rather small (Figure 2A), but that the effect on growth cone morphology is much larger upon mutation of both A and B interfaces, please speculate about possible models that might explain these somewhat divergent observations.

This is an interesting point. We agreed that, as presented in Figure 2A, mutations A and B make somewhat subtle, although significant, changes to the interaction propensities of Sema1b and PlexA measured by FRET at the surface of cells. This measurement is useful in that it does indicate that ligand – receptor interactions occur at the cell surface and that rather than being purely random these interactions are to a significant extent specific to the A and B binding sites. That these specific interactions are the ones that are important for the biology of the system is revealed by their effect on growth cone morphology.

Dear Yvonne,

Thank you for submitting your revised manuscript to the EMBO Journal. Your study has now been re-reviewed by the three referees and their comments are included below.

The referees still have some remaining concerns that should be resolved. I have looked at the points and I find them valid and they should be doable. Let me know if we need to discuss them further - I am happy to do so.

When you resubmit your revised version will you also take care of the following points:

- There is a callout for a Table S1 but there is no such table.
- The figures are very tightly cropped to the edges of the files maybe better to leave a bit more space to avoid problems when the figures are published.
- Are some of the images used in Fig 2B the same as shown in Fig EV3E? If so would you please add a sentence to the figure legend to reflect this.
- We don't permit data not shown (see p. 18) - see also guide to authors. I think you can simply remove this and leave the sentence as "All mutant Sema1b proteins were secreted at similar levels to the wild-type protein"
- There is a "Table EV1" in the manuscript. This should be renamed "Table 1"
- The legend for panel O in Fig EV2 is missing
- Our publisher has done their pre-publication check on your manuscript. We have uploaded the manuscript file - called Wiley Checked. Please take a look at the word file and the comments regarding the figure legends and respond to the issues. Please mark the changes - just makes it easier for me to see what has changed

with best wishes

Karin

Karin Dumstrei, PhD
Senior Editor
The EMBO Journal

- a point-by-point response to the referees' comments, with a detailed description of the changes made (as a word file).

- a word file of the manuscript text.

- individual production quality figure files (one file per figure)

- a complete author checklist, which you can download from our author guidelines (<https://www.embopress.org/page/journal/14602075/authorguide>).

- Expanded View files (replacing Supplementary Information)

Further information is available in our Guide For Authors:

The revision must be submitted online within 90 days; please click on the link below to submit the revision online before 11th Mar 2020.

Link Not Available

Referee #1:

The authors improved the description of experimental procedures and statistical analysis but several major points remain to be addressed:

- 1) The results in Figs. 3 and EV3G-K allow conclusions about the Sema1b and Sema6A mutants only if their cell surface expression is comparable and they do not affect the surface levels of plexins. The mutations introduced into Sema1b and Sema6A could affect their trafficking to the cell surface or indirectly the subcellular localization of plexins. A reduced surface expression of mutA+B has the same effect in these assays as abolishing the cis interaction. The authors claim that trafficking is not affected but do not explain how this was verified or show these results. The signals for PlexA-mClover in Fig. EV3G-K are concentrated in the perinuclear region suggesting that a considerable proportion of the protein is localized to intracellular compartments. The authors added images with the average fluorescence intensity for EGFP in Fig. 3B and mClover in Fig. EV3G-K but did not quantify it and do not explain how the average intensity was calculated. This

average intensity provides information only about the overall expression level but not the expression at the cell surface. The surface expression of both plexins and semaphorins has to be quantified by staining non-permeabilized cells with antibodies against an extracellular epitope (see e.g. Haklai-Topper et al., 2010).

Sema6A-EGFP intensity in Fig. 3B is shown only after growth cone collapse but the surface expression has to be determined before addition of ligand. In Fig. EV1 "the confocal images were taken from the membrane, thus potential changes in trafficking should not affect the monomeric or dimeric state on the cell surface". However, confocal microscopy does not allow to selectively image the plasma membrane.

2) Controls with vehicle only have to be included for the collapse assays in Fig. 3 not because the "analysis would even be stronger" but simply because it is good scientific practice to include an essential control (Kapfhammer et al., Nature Prot. 2007).

3) The dose-response measurements are essential for a complete characterization of the mutants and "It is good practice to obtain dose-response curves by applying increasing amounts of test protein" (Kapfhammer et al., Nat. Protoc. 2007). Haklai-Topper et al. (2010) report that 1.5 nM Sema6A-Fc induce the collapse of about 10% of the growth cones from wild type neurons compared to 25% from Sema6A knockout neurons. A 75 nM about 70% of wild type neurons and 80% of Sema6A knockout neurons are collapsed. Without testing different ligand concentrations it is not possible to decide whether mutation of site A and B are comparable to wild type constructs also at higher Sema6A concentrations. The authors claim that it is not feasible to perform these experiments but similar experiments can be found in numerous publications (e.g. Haklai-Topper et al., EMBO J. 2010; Suto et al., Neuron 2007; Takahashi et al., Cell 1999; Tawarayama et al., J. Neurosci. 2010; Xu et al., J. Neurosci. 2000). It would be sufficient and feasible to perform dose-response measurements with Sema6A using COS7 cells and sensory neurons since the required concentration is almost three orders of magnitude lower than that for Sema1b (75 nM compared to 5.8 μ M).

4) The authors included different images to improve the growth cone morphology matrix in Fig. 3 but the difference between categories 1 to 4 is still not obvious. Surprisingly, the growth cone included as category 3 in the revised manuscript appears to be the same as the example for category 2 in the previous version of the manuscript, which raises the question how reliable the assignment to these categories is. Why are 8 categories quantified to detect subtle changes but only the categories uncollapsed (1-4) and fully collapsed (5-8) compared for statistical analysis?

Minor points:

1) Fig. 3: Does "Totally, 20-40 growth cones were analyzed per condition" mean that 20-40 growth cones were analyzed for each of the n=3 experiments or 20-40 growth cones in total for all experiments?

2) Fig. EV4B: It is not mentioned if the intensity is shown of mClover or mRuby2.

Referee #2:

The authors have addressed most of my previous concerns. I now recommend the publication of this manuscript in the EMBO journal.

Referee #3:

I would like to thank the authors for addressing most of my concerns. However, the response to my first point is unsatisfactory.

The authors replied:

"As discussed in our response to Reviewer 2, given the low resolution of the 2:2 complex... we think that the resolution of the 2:2 complex (4.8 Å) is not sufficient to provide exact..."

Table S1 has information about the structures determined at resolutions ranging from 1.96 to 3.6 Å resolution. Where is this 4.8 Å resolution structure?

Please provide details about the molecular interfaces based on the structures shown in Table S1. Even at 3.6 Å resolution it should be possible to do this analysis.

Manuscript: Manuscript EMBOJ-2019-102926

Title: Structural basis of semaphorin-plexin *cis* interaction

Authors: Daniel Rozbesky, Marieke G. Verhagen, Dimple Karia, Gergely N. Nagy, Luis Alvarez, Ross A. Robinson, Karl Harlos, Sergi Padilla-Parra, R. Jeroen Pasterkamp, and E. Yvonne Jones

Referee #1:

The authors improved the description of experimental procedures and statistical analysis but several major points remain to be addressed:

1) The results in Figs. 3 and EV3G-K allow conclusions about the Sema1b and Sema6A mutants only if their cell surface expression is comparable and they do not affect the surface levels of plexins. The mutations introduced into Sema1b and Sema6A could affect their trafficking to the cell surface or indirectly the subcellular localization of plexins. A reduced surface expression of mutA+B has the same effect in these assays as abolishing the *cis* interaction. The authors claim that trafficking is not affected but do not explain how this was verified or show these results. The signals for PlexA-mClover in Fig. EV3G-K are concentrated in the perinuclear region suggesting that a considerable proportion of the protein is localized to intracellular compartments. The authors added images with the average fluorescence intensity for EGFP in Fig. 3B and mClover in Fig. EV3G-K but did not quantify it and do not explain how the average intensity was calculated. This average intensity provides information only about the overall expression level but not the expression at the cell surface. The surface expression of both plexins and semaphorins has to be quantified by staining non-permeabilized cells with antibodies against an extracellular epitope (see e.g. Haklai-Topper et al., 2010).

Sema6A-EGFP intensity in Fig. 3B is shown only after growth cone collapse but the surface expression has to be determined before addition of ligand. In Fig. EV1 "the confocal images were taken from the membrane, thus potential changes in trafficking should not affect the monomeric or dimeric state on the cell surface". However, confocal microscopy does not allow to selectively image the plasma membrane.

We thank the reviewer for raising this point. In response, we have carried out further work to strengthen our conclusion that all semaphorins and their mutants are expressed at comparable levels on the cell surface. Firstly, we have performed additional Total Internal Reflection (TIRF) microscopy experiments to measure fluorescence intensities on the cell surface. The TIRF microscopy provides much better Z resolution than confocal microscopy and allows us to selectively focus on the membrane, as the evanescent wave for the wavelengths utilized was calculated to be 130 nm above the coverslip. In our experiments, we observed comparable average intensities for all constructs indicating that Sema1b or Sema6A wild-type and all their mutants are expressed at similar levels on the cell surface (Page 9, 13, Fig EV3E-F, EV4C-D).

Furthermore, we have now added more controls for our FRET-FLIM experiments. In particular, we have provided a plot showing that calculated average lifetimes in our FRET-FLIM experiments are independent of donor/acceptor (Plexin/semaphorin) fluorescence intensity ratio (and therefore protein expression on the membrane) as shown by the random distribution of the PlexA-mClover/Sema1b-mRuby2 intensity ratio against the average lifetime (Fig EV3B).

2) Controls with vehicle only have to be included for the collapse assays in Fig. 3 not because the "analysis would even be stronger" but simply because it is good scientific practice to include an essential control (Kapfhammer et al., Nature Prot. 2007).

The reviewer repeatedly points to an elegant paper by Kapfhammer and colleagues for the use of vehicle controls in growth cone collapse assays. We acknowledge the relevance of such controls and use them in all our studies for assessing the effects of (wild type) axon guidance cues on growth cones (e.g. Van Battum et al., 2014, Nature Communications; Kong et al., 2016, Neuron). However, in the current study the experimental paradigm we use relies on comparison of growth cone collapse by Sema6A-Fc in control transfected Sema6A KO DRG neurons to the same neurons transfected with a full-length rescue *Sema6A* construct. As such we had not included a vehicle control, partly inspired by the study of Haklai-Topper published in EMBO journal in which no vehicle control was used for similar experiments (but instead a 'no treatment condition'). However, to address the reviewer's concern we have added a control treatment group for the EGFP-transfected *Sema6A* knockout neurons in our pilot study (0 nM) (Figure EV5). This now enables the reader even better to compare the 'uncollapsed' state of control transfected neurons (0 nM) to full growth cone collapse by Sema6A-Fc (75 nM; in control transfected neurons). This difference is the basis of our growth cone analysis (collapse response of EGFP transfected (maximal collapse response to Sema6A-Fc) versus control treatment (baseline level of collapse in control growth cones)).

3) The dose-response measurements are essential for a complete characterization of the mutants and "It is good practice to obtain dose-response curves by applying increasing amounts of test protein" (Kapfhammer et al., Nat. Protoc. 2007). Haklai-Topper et al. (2010) report that 1.5 nM Sema6A-Fc induce the collapse of about 10% of the growth cones from wild type neurons compared to 25% from Sema6A knockout neurons. A 75 nM about 70% of wild type neurons and 80% of Sema6A knockout neurons are collapsed. Without testing different ligand concentrations it is not possible to decide whether mutation of site A and B are comparable to wild type constructs also at higher Sema6A concentrations. The authors claim that it is not feasible to perform these experiments but similar experiments can be found in numerous publications (e.g. Haklai-Topper et al., EMBO J. 2010; Suto et al., Neuron 2007; Takahashi et al., Cell 1999; Tawarayama et al., J. Neurosci. 2010; Xu et al., J. Neurosci. 2000). It would be sufficient and feasible to perform dose-response measurements with Sema6A using COS7 cells and sensory neurons since the required concentration is almost three orders of magnitude lower than that for Sema1b (75 nM compared to 5.8 μ M).

In his/her original report the reviewer states that "*conclusions that can be drawn from the collapse assays are limited because only a single ligand concentration is used. It has been show that Sema6A can induce the collapse of sensory growth cones at high concentrations (Xu et al., 2000), which were not tested*".

Careful examination of the study by Xu et al. (2000) indicates that the highest concentration used by these authors is around 50 nM (50.000 pM), which is lower/similar to the amount of protein used in our study (75 nM). This amount of protein is also in line with the study of Haklai-Topper, which uses 75 nM Sema6A-Fc as the highest concentration (e.g. Figure 3C of their paper). Our experimental setup is based on the previous work by Haklai-Topper, namely that 75 nM of Sema6A-Fc induces robust collapse of Sema6A KO mouse DRG growth cones. Control transfection with EGFP-pCAG of Sema6A KO DRG neurons and treatment with Sema6A-Fc confirms this observation (Figure 3, this study). As predicted by the previous experiments of Yaron and colleagues, transfection of wild-type Sema6A-EGFP into mouse DRG neurons derived from Sema6A KO mouse partly reverses this collapse effect.

This setup and difference in collapse response is subsequently used to show that transfection of *Sema6A*-mutantA+B fails to rescue, whilst single mutants do. Although we feel that applying over 75 nM of *Sema6A*-Fc would not be physiologically meaningful (as it was also not done by our colleagues in the field: Yaron and Luo labs), we attempted to address the reviewer's point further by adding pilot data on the responses of *Sema6A* KO DRG growth cones at lower concentrations (Fig EV5) (at which the *Sema6A* collapse responses between WT and KO growth cones are somewhat larger than those reported at 75 nM by Haklai-Topper et al. (see paper Haklai-Topper et al. et al., EMBO J 2010)).

As outlined in our original rebuttal, we respectfully submit that the dose-response measurements are not feasible for our study. Referee 1 initially asked us to perform the dose-response measurements in order to calculate EC50 and pointed us to a number of publications (Haklai-Topper et al., EMBO J. 2010; Suto et al., Neuron 2007; Takahashi et al., Cell 1999; Tawarayama et al., J. Neurosci. 2010; Xu et al., J. Neurosci. 2000). However, only 2 out of those 5 publications show EC50s, and in neither is the analysis rigorous.

First, the number of experimental points (usually 4-5) in all those publications is not sufficient for sigmoidal fitting. Indeed, there is no real mathematical fitting; the points are simply interconnected. We believe that this approach cannot be used for the determination of EC50. As we said in our first rebuttal, we would need to generate the dose-response curve for three mutants (A, B, A+B) and one wild-type protein of each *Sema1b* and *Sema6A* at least in triplicates. That would require measurement and analysis of at least 360 different conditions (4x2x15x3). Also, this approach would necessitate the sacrifice of a large number of mice, and we do not think that a case could be made to gain permission from the ethical committee.

However, to also address this point and to strengthen our conclusions from the growth cone collapse assay in Figure 3, we now provide pilot data for the different rescue constructs at two additional ligand concentrations, 1 and 10 nM. We do not find significant growth cone collapse responses at 1 nM at any of the conditions. At 10 nM, the ratio uncollapsed:collapsed KO growth cones is shifted in the *Sema6A*-mutA+B condition, indicating that at this concentration the double mutant fails to completely rescue the lack of endogenous *Sema6A* (as observed at 75 nM). We have added these data as Supplemental data to Figure EV5.

4) The authors included different images to improve the growth cone morphology matrix in Fig. 3 but the difference between categories 1 to 4 is still not obvious. Surprisingly, the growth cone included as category 3 in the revised manuscript appears to be the same as the example for category 2 in the previous version of the manuscript, which raises the question how reliable the assignment to these categories is. Why are 8 categories quantified to detect subtle changes but only the categories uncollapsed (1-4) and fully collapsed (5-8) compared for statistical analysis?

We apologize for this mistake. While restructuring the matrix in Figure 3 in response to a comment of this reviewer, the growth cone representing category 3 was mistakenly shifted in the Figure. We have corrected this mistake and have further improved the matrix. We would like to indicate that in our revised manuscript we have addressed all the original concerns raised by this reviewer regarding the matrix (e.g. regarding an explanation of the categories, improvement of the legends etc). Again we would like to emphasize that both the morphological and statistical analysis of growth cone collapse as applied in this study have been approved by and published in several high profile journals. Nevertheless, we have

reworded the text to further clarify how and why the different categories were used (p. 25-26).

Minor points:

1) Fig. 3: Does "Totally, 20-40 growth cones were analyzed per condition" mean that 20-40 growth cones were analyzed for each of the n=3 experiments or 20-40 growth cones in total for all experiments?

We have clarified this point in the text (p. 32-33).

2) Fig. EV4B: It is not mentioned if the intensity is shown of mClover or mRuby2.

Thank you, we have now clarified this point in the Figure legend (p. 36).

Referee #2:

The authors have addressed most of my previous concerns. I now recommend the publication of this manuscript in the EMBO journal.

We thank the reviewer for supporting publication of our manuscript in EMBO journal.

Referee #3:

I would like to thank the authors for addressing most of my concerns. However, the response to my first point is unsatisfactory.

The authors replied:

"As discussed in our response to Reviewer 2, given the low resolution of the 2:2 complex... we think that the resolution of the 2:2 complex (4.8 Å) is not sufficient to provide exact..."

Table S1 has information about the structures determined at resolutions ranging from 1.96 to 3.6 Å resolution. Where is this 4.8 Å resolution structure?

Please provide details about the molecular interfaces based on the structures shown in Table S1. Even at 3.6 Å resolution it should be possible to do this analysis.

We believe that the reviewer's comment may be based on a partial misunderstanding. In particular, the resolution of the 2:2 complex in Table S1 (now Table 1 in response to Editor's comments below) is really 4.8 Å, and as we said before, we are of the view that this is not sufficient to provide details about the molecular interfaces. Table 1 in this manuscript does not show the resolutions ranging from 1.96 to 3.6 Å resolution as Referee 3 suggests. However, the structures with resolutions ranging from 1.96 to 3.6 Å have been shown in our Nat Communications manuscript, which was requested by the editor to provide together with our EMBO J manuscript. Thus, we think that the misunderstanding is based on confusion of Table S1s between our EMBO J and Nat Communications manuscript.

Editor comments:

- There is a callout for a Table S1 but there is no such table.

Thank you for this point. We apologise, Table S1 in our original manuscript was renamed Table EV1 in the first revised manuscript and we failed to change the callout in the Methods. In accordance with the request to re-name Table EV1 (see below) we have corrected the call out of Table S1 to Table 1.

- The figures are very tightly cropped to the edges of the files maybe better to leave a bit more space to avoid problems when the figures are published.

We have now amended the figures.

- Are some of the images used in Fig 2B the same as shown in Fig EV3E? If so would you please add a sentence to the figure legend to reflect this.

Thank you. Yes, Fig 2B shows representative 2D class averages of PlexA_{ecto} while Fig EV3E (now Fig EV3H) shows all 2D class averages. We have added a sentence in the Fig 2B legend to address this point.

- We don't permit data not shown (see p. 18) - see also guide to authors. I think you can simply remove this and leave the sentence as "All mutant Sema1b proteins were secreted at similar levels to the wild-type protein"

Thank you. We have removed "data not shown" from that sentence.

- There is a "Table EV1" in the manuscript. This should be renamed "Table 1"

We have renamed Table EV1 to Table 1.

- The legend for panel O in Fig EV2 is missing

Apologies, we have provided a legend.

- Our publisher has done their pre-publication check on your manuscript. We have uploaded the manuscript file - called Wiley Checked. Please take a look at the word file and the comments regarding the figure legends and respond to the issues. Please mark the changes - just makes it easier for me to see what has changed

We have responded to all the issues raised in the Wiley Checked manuscript.

Dear Yvonne and Jeroen,

Thanks for submitting your revised manuscript to The EMBO Journal. Your revision has now been re-reviewed by the two referees and I am pleased to let you know that we will accept the manuscript for publication in the EMBO Journal.

Before sending you the official acceptance letter there are just a few last min things to sort out.

- Can you make sure to describe how the average fluorescence intensity was done (Referee #1)
- Regarding Fig. EV5 (referee #1) - I have no strong feelings either way. I don't think it harms leaving EV5 in but will leave that decision up to you.
- I have asked our publisher to do their pre-publication check on this manuscript but have not received their comments yet. I will pass them on as soon as I get them. Please don't submit the revision before you receive the comments from me.
- We include a synopsis of the paper (see <http://emboj.embopress.org/>). Please provide me with a general summary statement and 3-5 bullet points that capture the key findings of the paper.
- We also need a summary figure for the synopsis. The size should be 550 wide by 400 high (pixels).

You can use the link below to upload the revised manuscript.

That should be all!

Congratulations on a nice study.

With best wishes

Karin

Karin Dumstrei, PhD
Senior Editor
The EMBO Journal

Further information is available in our Guide For Authors:

The revision must be submitted online within 90 days; please click on the link below to submit the revision online before 8th Jul 2020.

Link Not Available

Referee #1:

The authors added controls to confirm that the surface expression of Sema6A mutants is comparable in COS-7 cells but these controls were not performed for the experiments shown in Fig. 3. The calculation of average fluorescence intensity is not described in sufficient detail. The only information in Methods is that the analysis was done using ImageJ.

The authors included results from a single experiment (Fig. EV5) that was done as a pilot study to test the effect of different Sema6A-Fc concentration and quantified as percent collapsed growth cones. Fig. EV5 does not allow any conclusions and should not be included because controls with mock (GFP)-transfected cells are missing for 1 and 10 nM Sema6A-Fc.

The collapse of growth cones was quantified in Fig. 3 using 8 categories. The authors refer to two previous publications in "high profile journals" that "approved" this analysis (Kong et al., 2016, van Erp et al., 2015). Kong et al. used 10 categories and van Erp quantified the percentage of collapsed growth cones. The publication in "high profile journals" does not absolve the authors from the obligation to properly describe the assay. The authors now corrected the figure and added a description of their categories.

Referee #3:

I'm satisfied with the author's responses to my comments on their original submission. I recommend publication as is.

Manuscript: Manuscript EMBOJ-2019-102926

Title: Structural basis of semaphorin-plexin *cis* interaction

Authors: Daniel Rozbesky, Marieke G. Verhagen, Dimple Karia, Gergely N. Nagy, Luis Alvarez, Ross A. Robinson, Karl Harlos, Sergi Padilla-Parra, R. Jeroen Pasterkamp, and E. Yvonne Jones

Referee #1:

The authors added controls to confirm that the surface expression of Sema6A mutants is comparable in COS-7 cells but these controls were not performed for the experiments shown in Fig. 3. The calculation of average fluorescence intensity is not described in sufficient detail. The only information in Methods is that the analysis was done using ImageJ. The authors included results from a single experiment (Fig. EV5) that was done as a pilot study to test the effect of different Sema6A-Fc concentration and quantified as percent collapsed growth cones. Fig. EV5 does not allow any conclusions and should not be included because controls with mock (GFP)-transfected cells are missing for 1 and 10 nM Sema6A-Fc. The collapse of growth cones was quantified in Fig. 3 using 8 categories. The authors refer to two previous publications in "high profile journals" that "approved" this analysis (Kong et al., 2016, van Erp et al., 2015). Kong et al. used 10 categories and van Erp quantified the percentage of collapsed growth cones. The publication in "high profile journals" does not absolve the authors from the obligation to properly describe the assay. The authors now corrected the figure and added a description of their categories.

We thank the reviewer for raising this point. In response, we have described the calculation of average fluorescence intensity (Page 23). Regarding EV5, we have decided to keep EV5 in our manuscript.

Referee #3:

I'm satisfied with the author's responses to my comments on their original submission. I recommend publication as is.

We thank the reviewer for supporting publication of our manuscript in EMBO journal.

Editor comments:

- Mismatch between legend and figure panels in EV3

Thank you for this point. We apologise, we have now corrected the legend.

Dear Yvonne,

Thank you for submitting your revised MS to The EMBO Journal. I have now had a careful look at the revised version and all looks good.

I am therefore very pleased to accept the MS for publication here.

Congratulations on a nice study

with best wishes

Karin

Karin Dumstrei, PhD
Senior Editor
The EMBO Journal

Please note that it is EMBO Journal policy for the transcript of the editorial process (containing referee reports and your response letter) to be published as an online supplement to each paper. If you do NOT want this, you will need to inform the Editorial Office via email immediately. More information is available here: http://emboj.embopress.org/about#Transparent_Process

Your manuscript will be processed for publication in the journal by EMBO Press. Manuscripts in the PDF and electronic editions of The EMBO Journal will be copy edited, and you will be provided with page proofs prior to publication. Please note that supplementary information is not included in the proofs.

Should you be planning a Press Release on your article, please get in contact with embojournal@wiley.com as early as possible, in order to coordinate publication and release dates.

If you have any questions, please do not hesitate to call or email the Editorial Office. Thank you for your contribution to The EMBO Journal.

EMBO Press encourages all authors and reviewers to associate an Open Researcher and Contributor Identifier (ORCID) to their account. ORCID is a community-based initiative that provides an open, non-proprietary and transparent registry of unique identifiers to help disambiguate research contributions.

Currently, our records indicate that the ORCID for your account is 0000-0002-3834-1893.

Please click the link below to modify this ORCID:
Link Not Available

** Click here to be directed to your login page: <http://emboj.msubmit.net>

Corresponding Author Name: Yvonne Jones

Journal Submitted to: EMBO J

Manuscript Number: EMBOJ-2019-102926